# Synthetic phage-based approach for sensitive and specific detection of *Escherichia coli* O157
Azumi Tamura[1,2], Aa Haeruman Azam[1], Tomohiro Nakamura[1], Kenichi Lee [ID][3], Sunao Iyoda[3], Kohei Kondo[1], Shinjiro Ojima[1], Kotaro Chihara [ID][1], Wakana Yamashita[1,4], Longzhu Cui [ID][5], Yukihiro Akeda[3], Koichi Watashi[1], Yoshimasa Takahashi [ID][1,4], Hiroshi Yotsuyanagi[2] & Kotaro Kiga [ID][1,5] ✉

*Escherichia coli* O157 can cause foodborne outbreaks, with infection leading to severe disease such as hemolytic-uremic syndrome. Although phage-based detection methods for *E. coli* O157 are being explored, research on their specificity with clinical isolates is lacking. Here, we describe an in vitro assembly-based synthesis of vB_Eco4M-7, an O157 antigen-specific phage with a 68-kb genome, and its use as a proof of concept for *E. coli* O157 detection. Linking the detection tag to the C-terminus of the tail fiber protein, gp27 produces the greatest detection sensitivity of the 20 insertions sites tested. The constructed phage detects all 53 diverse clinical isolates of *E. coli* O157, clearly distinguishing them from 35 clinical isolates of non-O157 Shiga toxin-producing *E. coli*. Our efficient phage synthesis methods can be applied to other pathogenic bacteria for a variety of applications, including phage-based detection and phage therapy.

*Escherichia coli* O157 has emerged as a major food- and water-borne pathogen, causing outbreaks and sporadic cases of illness worldwide[1–3]. It is one of the most recognized and frequently associated serotypes of Shiga-toxin-producing *E. coli* (STEC), causing an estimated 63,000 STEC O157 infections annually in the United States, resulting in ~2000 hospitalizations and 20 deaths[4]. The production of Shiga toxins by this strain can lead to life-threatening complications, including hemorrhagic colitis and hemolytic-uremic syndrome[5,6]. The susceptibility of infants and children to *E. coli* O157 infection is of particular concern, as they are more prone to developing hemolytic-uremic syndrome[7]. Given the association between antibiotic use and hemolytic-uremic syndrome development, there has been increased focus on prioritizing the prevention of primary infections[5,8,9]. Considering the potential for even low levels of *E. coli* O157 to cause infection[10], detecting this pathogen should be rapid, sensitive, and specific. In particular, strains belonging to clade 8 of the O157 serotype have a higher association with severe patient outcomes among 9 distinct clades[11]. Therefore, early and accurate detection of *E. coli* O157 strains belonging to this clade is critical[11].

Traditional culture methods such as selective agar plating and biochemical testing are widely used to identify and isolate *E. coli* O157[5,12,13].

These detection methods specifically target viable bacteria that have the potential to cause illness, thereby ensuring high accuracy and reliability. However, identification requires multiple steps; 3–7 days are required to obtain negative results, and an additional 2 days or longer to conduct biochemical or molecular tests to obtain positive results[14,15]. Other methods, including molecular and immunological approaches, have shortened detection times[16]. PCR-based methods provide rapid and sensitive detection; however, they require specialized equipment and expertise, are more costly than culture-based methods, and cannot distinguish between live and dead bacteria[17–19].

Bacteriophages (also known as phages) have gained attention for their utility in specific bacterial detection because of their high specificity and ability to infect only living bacteria[19]. A recent study has shown high sensitivity, specificity, and accuracy of phage-based methods for detecting *E. coli*, *Enterococcus* spp., and *Klebsiella* spp. in urine samples[20]. Phage-based methods encompass various approaches to bacterial detection. These include detection methods based on intracellular substances such as adenosine triphosphate released during phage-induced bacterial cell lysis[21], phage particle adsorption on bacterial cell walls[22], and fluorescently labeled

[1]Research Center for Drug and Vaccine Development, National Institute of Infectious Diseases, Shinjuku-ku, Tokyo, Japan. [2]Division of Infectious Diseases, Advanced Clinical Research Center, The Institute of Medical Science, The University of Tokyo, Minato-ku, Tokyo, Japan. [3]Department of Bacteriology I, National Institute of Infectious Diseases, Shinjuku-ku, Tokyo, Japan. [4]Department of Life Science and Medical Bioscience, Waseda University, Shinjuku-ku, Tokyo, Japan. [5]Division of Bacteriology, Department of Infection and Immunity, School of Medicine, Jichi Medical University, Shimotsuke-shi, Tochigi, Japan. ✉e-mail: k-kiga@niid.go.jp

phages[23,24]. Recently, bacterial chromosomal islands integrated into phages have been harnessed for detection and identification of *E. coli* and *Staphylococcus aureus*[25]. Moreover, pathogen detection has been facilitated using specific fusion phage proteins selected from phage display libraries[26].

Genetic modifications of phages, such as incorporating reporter genes, have been reported to enhance the detection sensitivity of phage-based detection methods[20,27,28]. For example, the luciferase gene has been introduced into a temperate *E. coli* O157 phage, phiV10, via homologous recombination generating chimera phages for bacterial detection[29,30]. Moreover, a virulent phage, PP01, has been modified to carry green fluorescence protein for *E. coli* O157 detection[31–33]. However, the ability of these modified phages to detect various *E. coli* O157 strains, particularly clinical isolates, is unconfirmed. A novel virulent phage, vB_Eco4M-7, was recently reported[34]. Because of its rapid propagation and ability to effectively lyse various *E. coli* O157 strains, using this phage for pathogen detection in food is expected to enhance food safety measures.

Several strategies have been developed for phage genome engineering. Genome engineering techniques, such as those involving homologous recombination systems, are time consuming and necessitate an intensive screening process. They are primarily suitable for temperate phages that integrate their genome into the bacterial genome as prophages in bacteria[35,36]. Manipulating virulent phages, which have only a lytic cycle, is more complex; however, in vivo and ex vivo phage engineering methods have recently been developed[35]. In vivo phage engineering approaches include homologous recombination[37], recombineering[38], and clustered regularly interspaced short palindromic repeats/CRISPR-associated protein (CRISPR/Cas) systems for counter-selection[39,40]. Although these approaches have been extensively used, they have numerous limitations. These include the limited synthetic efficiency depending on the phage genome length, requirement for genetic manipulation of host bacteria, difficulties in genetic manipulation within the host bacterium arising from the phage genes that are toxic to the host bacterium, and inability to simultaneously edit multiple phage genes[35,41].

In contrast, ex vivo phage engineering techniques involve assembling and rebooting synthetic phage genomes, thereby enabling full synthesis and flexible phage design[35]. Synthetic phage genomes could be constructed using PCR-amplified or synthetic DNA fragments which could then be assembled with different approaches, such as yeast-artificial-chromosome-assisted assembly in yeast cells[42,43] and in vitro assembly, such as Gibson assembly[41,44]. Finally, the chimera phages could be generated using conventional transformation techniques[41–44], using L-form bacteria that lack a cell wall[45,46], or cell-free transcription-translation (TX-TL) systems[44,47–49]. In vitro DNA assembly represents a straightforward and rapid technique for phage engineering[41]. However, synthesizing phages with large genomes presents challenges, highlighting the necessity for enhancing the efficiency and applicability of phage synthesis methods.

Herein, we present an in vitro method for synthesizing an *E. coli* O157-specific phage with a 68-kb genome. The efficient phage-synthetic method enables a targeted phage to be used in a variety of applications, including therapy against drug-resistant bacteria. We conducted a proof of concept study using the phage to detect bacteria expressing its target receptor. Through a comparison of the detection tag insertion sites, we demonstrate a rapid and ultrasensitive *E. coli* O157 detection system based on synthetic tagged phages. The synthetic phage-based detection method could be extended to other pathogenic bacteria, allowing the simultaneous monitoring and detection of various pathogenic bacteria.

## Results

### *E. coli* O157 phage and its host specificity
To detect *E. coli* O157, we first searched for phages specifically infecting *E. coli* O157. We compared the genomes of T1–T7, which infect *E. coli*, and those of SP15, PP01, and vB_Eco4M-7 (GenBank: MN176217), which infect *E. coli* O157[34,50,51] (Fig. 1a). The O157 phage vB_Eco4M-7 (hereafter referred to as O157_vB) was relatively similar to the T3 and T7 phages, SP15 was similar to the T5 phage, and PP01 was similar to the T2, T4, and T6

phages (Fig. 1a). Subsequently, we assessed the bactericidal activity against eight *E. coli* strains (Fig. 1b). Both O157_vB and PP01 showed lytic activity only against *E. coli* O157 (Fig. 1b). In contrast, T1–T7 effectively lysed *E. coli* K-12 and B strains but not serotype O157, and SP15 showed strong lytic activity against *E. coli* K-12 and B strains as well as O157 (Fig. 1b). To determine the host range of O157_vB and PP01, the phage susceptibility to O157 and non-O157 STEC clinical isolates was tested using spot assay (Fig. 1c). O157_vB infected all the tested *E. coli* O157 isolates, albeit with variable lytic efficiency, as indicated by the efficiency of plating (EOP) values (Fig. 1c). In contrast, O157_vB did not infect any of the 35 non-O157 STEC isolates belonging to 11 different O-serotypes. PP01 is an *E. coli* O157-specific phage[51]; however, it also infects other STECs, such as O145 and O5 (Fig. 1c). O157_vB infects *E. coli* O157 more specifically than PP01, making it suitable for the specific detection of *E. coli* O157.

### Prediction of O157_vB host receptor
Furthermore, we isolated and analyzed phage-resistant *E. coli* O157 mutants to predict the host receptor for O157_vB (Supplementary Fig. 1a, b and Supplementary Table 1). The results were compared with those of PP01 and SP15 (Fig. 2a). Mutations in enzymes involved in lipopolysaccharide synthesis[52–54] rendered *E. coli* O157 resistant to O157_vB (Fig. 2a and Supplementary Fig. 1a). In contrast, no such mutations were found in PP01 or SP15-resistant mutants (Fig. 2a). Similarly, O157_vB exhibited no lytic activity against the *rfbE*-deficient mutants unable to express the O157 antigen[55,56] and recovered infectivity by supplying an RfbE-expressing plasmid (Fig. 2b and Supplementary Figs. 1a and 2). Based on the specificity to *E. coli* O157 using O157 antigen as a receptor, we selected O157_vB for *E. coli* O157 detection.

### In vitro synthesis of *E. coli* O157 phage with a 68 kb genome
We next used in vitro assembly-based synthesis methods to reboot the O157_vB phages (Fig. 3a). First, we extracted the genomic DNA from O157_vB and obtained DNA fragments via PCR. PCR primers were designed such that adjacent DNA fragments overlapped. Using the overlaps, the fragments were assembled in vitro. The assembled products were then electroporated into *E. coli* HST08, which is not an O157_vB natural host. O157_vB phages were eventually rebooted from the assembled DNA inside the bacteria, and co-incubation with *E. coli* O157 enabled the phage propagation, which formed plaques on the plates.

Because the genome size of O157_vB is larger than 68 kb, which exceeds the typical range of 50 kb used in in vitro synthesis methods, we enhanced the efficiency of phage synthesis by considering several key factors. Gel purification was performed for the PCR fragments (Supplementary Fig. 3a). The DNA fragments used for the assembly had overlapping lengths of ~40 bp (Supplementary Fig. 3b). It was crucial to maintain the amount of DNA fragments required for assembly per reaction (30 μl) below 50 fmol (Supplementary Fig. 3c). Using these optimized methods, O157_vB was successfully synthesized and rebooted (Supplementary Fig. 4a, b). Obtaining the synthetic O157_vB from its genome took only a few days. In addition, the bactericidal activity of the synthetic O157_vB was equivalent to that of natural O157_vB, indicating that the synthetic O157_vB retained its functional properties (Supplementary Fig. 4c).

Next, we added an 11-amino acid peptide (HiBiT) tag[57] to the O157_vB protein for *E. coli* O157 detection. We selected the minor capsid protein (gp37) and bacteriolysis enzyme (gp64) as target proteins, whose gene functions are currently being elucidated, and added HiBiT tags to their C-termini (Fig. 3b). Similar to synthesizing the untagged O157_vB, the HiBiT-tagged O157_vB was synthesized by incorporating the HiBiT sequence into the PCR primers (Fig. 3b and Supplementary Fig. 5a, b). The killing efficiency of the tagged phages did not differ from that of the natural O157_vB (Fig. 3c).

### Detection of *E. coli* O157 using HiBiT-tagged O157_vB
Next, using the synthetic O157_vB, we performed *E. coli* O157 detection assays. HiBiT binds with high affinity to an 18-kDa subunit called

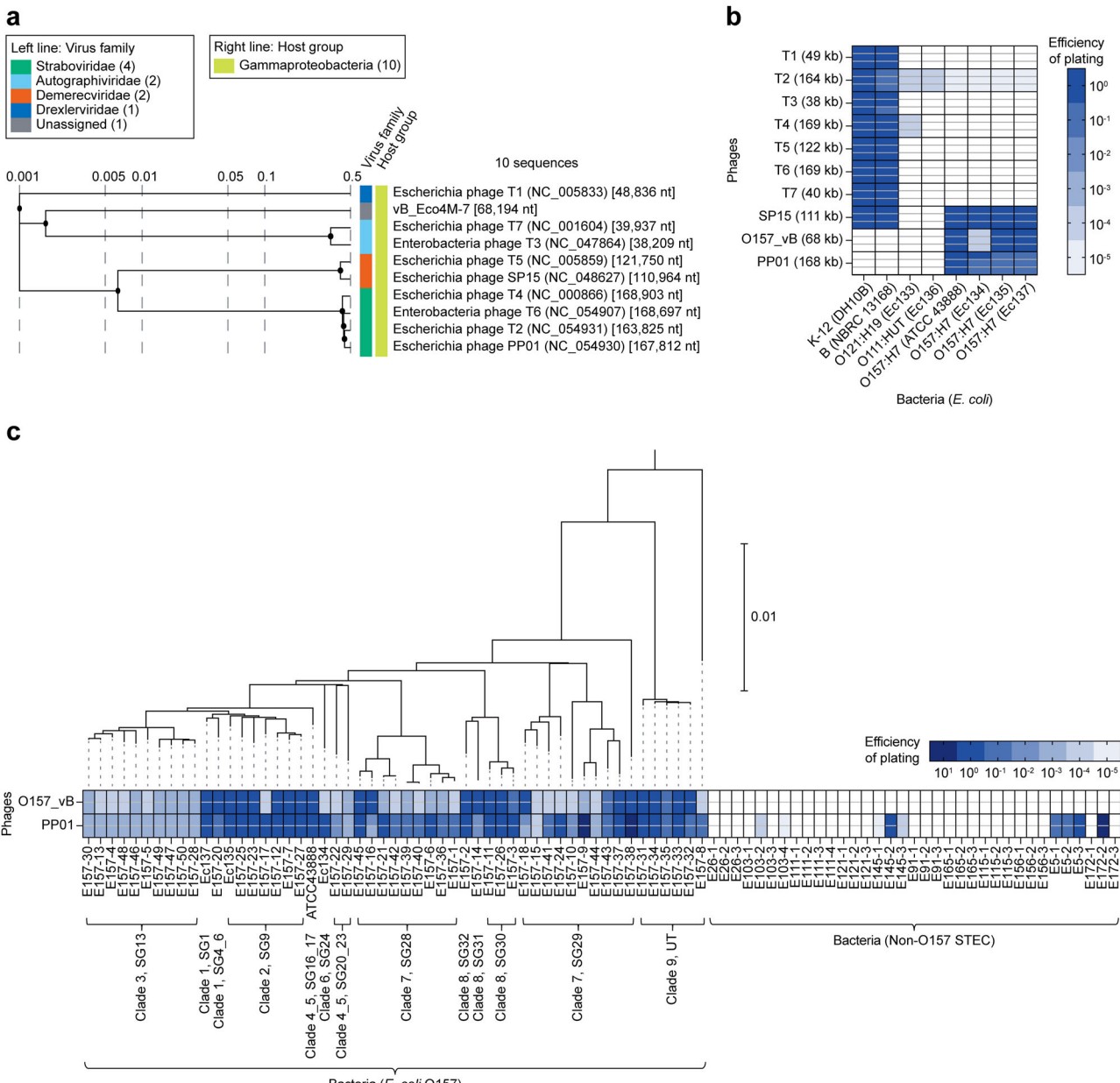

**Fig. 1 | E. coli O157 phage and its host specificity. a** Phage proteomic tree was generated using ViPTree based on whole-genome similarities computed via tBLASTx. *E. coli* O157-specific phage, vB_Eco4M-7 (O157_vB), was compared with coliphage T1–T7, SP15, and PP01. Colored lines indicate virus family on left and host group on right. Phage names are followed by RefSeq (NCBI Reference Sequences) accession number and genome length. **b** Phage lytic activities against 8 *E. coli* strains were measured via spot tests. Efficiency of plating (EOP) against each *E. coli* strain was calculated by comparing plaque counts between each and natural host. Three technical replicates were prepared. **c** Lytic activities of O157_vB and PP01 against clinical isolates of 53 O157 and 35 non-O157 Shiga toxin-producing *E. coli* (STEC), with maximum likelihood phylogeny of STEC O157 strains. Clinical isolates with names beginning with E157 were *E. coli* O157 strains, and numbers following E denote serotypes of *E. coli* strains. Spot tests were performed twice for each strain. Phylogenic tree was rooted in *E. coli* O26:H11 11368 (accession number AP010953). Scale bar represents substitution rate per site.

LgBiT, derived from NanoLuc, and the bound complex produces strong and quantifiable luminescence in the presence of a substrate[57,58]. After a 2-h incubation of $10^7$ colony-forming units (CFU)/ml bacteria with $10^7$ plaque-forming units (PFU)/ml O157_vB, the culture exhibited a strong luminescence signal, distinguishing it from non-O157 *E. coli* (Fig. 4). The lytic activity of O157_vB against *E. coli* O157 was confirmed using bacterial-reduction curves (Supplementary Fig. 6). Furthermore, O157_vB with the HiBiT tag at gp64 (lytic enzyme) exhibited ~10 times more luminescence than the O157_vB tagged at gp37 (minor capsid protein), indicating that the tag location affects detection sensitivity (Fig. 4).

### Various O157_vB phages with HiBiT on different proteins

Next, to examine the proteins suitable for HiBiT tagging, we synthesized various O157_vB phages using HiBiT tags with different proteins. We selected 20 proteins for comparison, including structural proteins such as capsid and tail proteins, lytic enzymes, proteins involved in DNA replication, and proteins with genes immediately after transcription start sites where high expression was expected (Table 1). When attempting to synthesize phages with tags, the insertion of a tag sequence into certain genes greatly reduced the plaque count. This was due to the introduction of a tag to the essential targeted genes, thereby inducing structural and functional alterations. Of the 20 synthesis attempts, 13 successfully generated O157_vB

**Fig. 2 | Prediction of O157_vB host receptor.**
**a** Mutations identified in phage-resistant bacteria. Phage-resistant mutants were obtained after over-night incubation of *E. coli* O157 (ATCC 43888) and *E. coli* O157 phages. Their whole genomes were compared with ATCC 43888 wildtype, and core SNPs were identified. **b** Lytic activity of O157_vB against *rfbE*-deficient *E. coli* O157 strains. Two wildtype strains (ATCC 43888 and Sakai), their *rfbE*-deficient mutants, and *rfbE*-deficient ATCC 43888 harboring RfbE-expressing plasmid (pGEM-T-Easy-*rfbE*) were tested. Three technical replicates were prepared.

**a**

| *E. coli* O157 | Mutation | | Gene | Product | Source |
|---|---|---|---|---|---|
| O157_vB-resistant | A→C | Val133Gly | *wbdO* | Glycosyltransferase | This study |
| | G→T | Ala83Asp | *wbdN* | Glycosyltransferase | This study |
| | A→C | Leu196Arg | *wbdN* | Glycosyltransferase | This study |
| | G→A | Arg265Cys | *wecA* | UDP-N-acetylglucosamine--undecaprenyl-phosphate N-acetylglucosamine phosphotransferase | This study |
| | G→GC | Ala106fs | *wzy* | O157 family O-antigen polymerase | This study |
| PP01-resistant | C→T | Gln76Ter | *ompC* | Porin OmpC | Azam *et al.* [50] |
| | A→T | Arg143Ter | – | Glycosyltransferase | [50] |
| SP15-resistant | C→T | Trp511Ter | *fhuA* | Ferrichrome porin FhuA | [50] |
| | G→A | Pro696Leu | – | DEAD/DEAH box helicase | [50] |

**b**

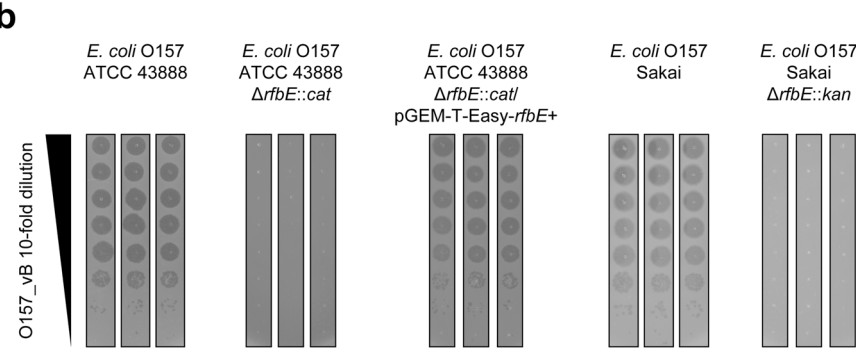

phages, which were then used in an *E. coli* O157 detection assay to compare their detection sensitivity (Table 1, Fig. 5a and Supplementary Fig. 7a, b). The amount of observed luminescence varied among the 13 phages, and O157_vB with a HiBiT tag at gp27 (O157_vB$_{HiBiT(gp27)}$), a tail fiber protein, showed the highest level of luminescence after a 2-h incubation with *E. coli* O157 (Fig. 5a). Statistical analysis using multi-group comparisons confirmed significant differences (Supplementary Table 2).

We then investigated the detection of small numbers of *E. coli* O157 using O157_vB$_{HiBiT(gp27)}$ (Fig. 5b, c). O157_vB$_{HiBiT(gp27)}$ detected ~$10^2$ CFU/ml (18 CFU/200 µl) *E. coli* O157 of a Shiga-toxin-deficient ATCC strain (ATCC 43888) and clinical isolates Ec134 and Ec135 after a 2-h co-incubation (Fig. 5b and Supplementary Table 3). *E. coli* O157 concentrations of $10^3$ CFU/ml or higher were linearly correlated with luminescence levels, whereas those of $10^2$ CFU/ml did not exhibit such correlation (Supplementary Fig. 8). Moreover, after a 5-h incubation of 1 CFU/ml (equivalent to 10 CFU/10 ml) *E. coli* O157 ATCC 43888 with O157_vB$_{HiBiT(gp27)}$ for 2 h led to successful detection (Fig. 5c and Supplementary Table 4).

### Detection of *E. coli* O157 clinical isolates

To assess the usefulness of HiBiT-tagged O157_vB for *E. coli* O157 detection, we next examined the detection of 53 clinical isolates of STEC O157 from 8 diverse clades, as well as 35 clinical isolates of non-O157 STEC across 11 different O-serotypes, using O157_vB$_{HiBiT(gp27)}$ (Fig. 6a). The luminescence levels of *E. coli* O157 differed among the strains but correlated very well with the phage infectivity (Fig. 6a, b). Specifically, all O157 isolates belonging to clade 3 were killed inefficiently by O157_vB and had relatively low luminescence levels, possibly due to anti-phage defense systems specific to clade 3, such as DarTG[59] and Zorya type I[60] (Figs. 1c, 6a and Supplementary Table 5). On the other hand, O157_vB efficiently killed and

detected O157 strains from clade 8, the proposed hypervirulent lineage (Figs. 1c and 6a). When non-O157 STEC strains were infected with O157_vB$_{HiBiT(gp27)}$, no sample emitted luminescence, indicating that the detection system was highly specific (Fig. 6a). A comparison of the luminescence levels of O157 and non-O157 STEC confirmed that the synthesized O157_vB$_{HiBiT(gp27)}$ could clearly distinguish these bacteria (Fig. 6c).

### Discussion

In this study, we examined the conditions for in vitro phage synthesis using the T7 phage and successfully synthesized O157_vB from a genome with a size exceeding 68 kb. To the best of our knowledge, this is the first successful in vitro assembly-based synthesis of an *E. coli* O157 phage. In vitro assembly-based synthesis offers greater flexibility than recombination-based techniques in manipulating phage genomes. This enables a straightforward construction and facilitates the alteration of phage host range[44]. This method also improves synthesis efficiency and eliminates the need for phage selection.

Typically, phage genomes are limited by the gene length that can be inserted into the capsid, making it challenging to increase phage genome length. However, introducing the HiBiT tag into the *E. coli* O157 phage was easily achieved. The HiBiT tag, which comprises only 11 amino acids, was of sufficient length to fit within the phage capsid. *E. coli* strain HST08, which is not a natural host of O157_vB, was used for phage reconstitution because of its high electroporation efficiency. Efficient reconstitution of O157_vB was achieved by rebooting it in HST08 and co-culture with *E. coli* O157 as the phage host. This method could enable the synthesis of various phages targeting bacteria closely related to *E. coli*.

We confirmed the high specificity of O157_vB in infecting *E. coli* O157. O157_vB did not infect non-O157 strains, including *E. coli* O157, with mutations in lipopolysaccharide synthesis enzymes, suggesting it uses the

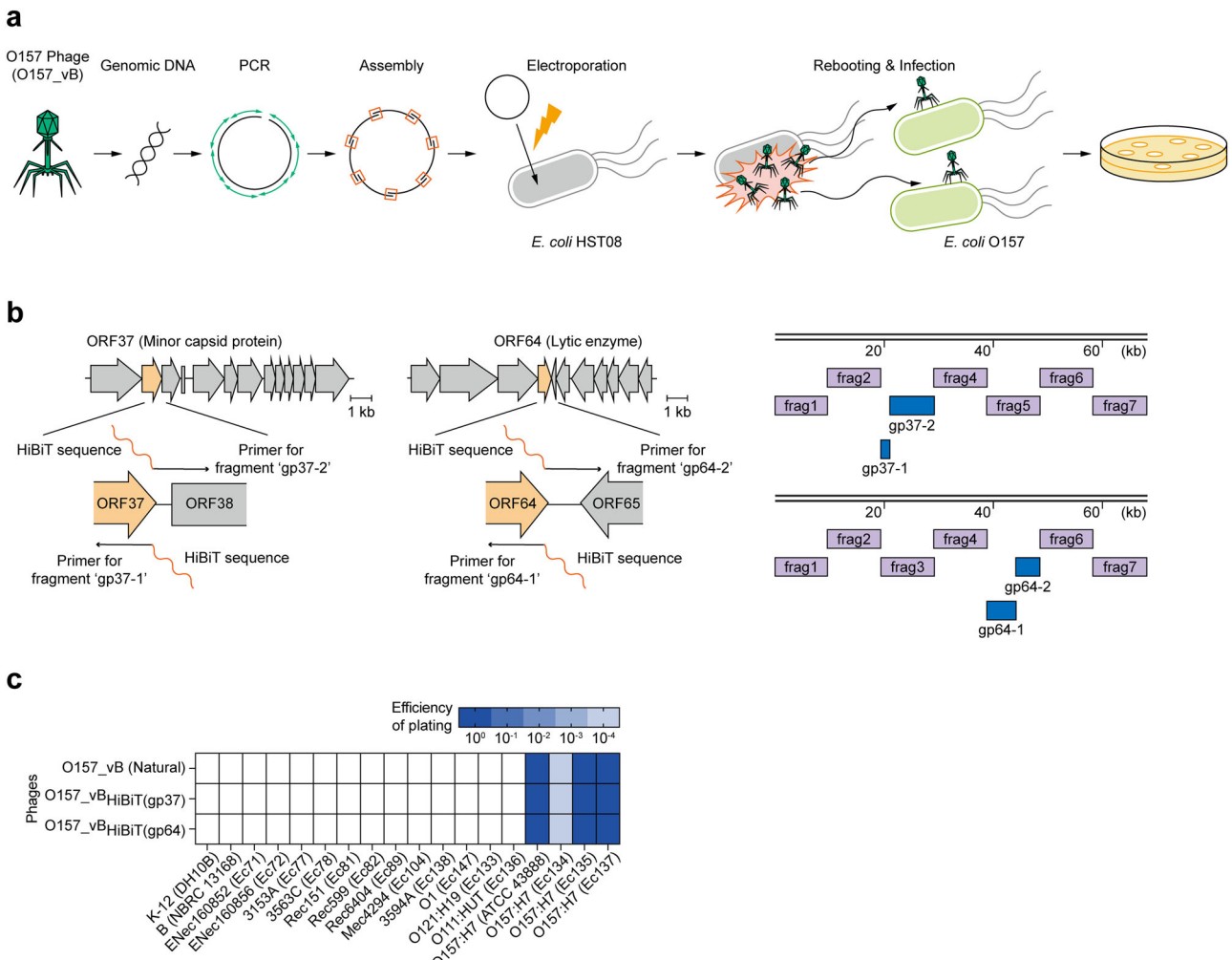

**Fig. 3 | In vitro synthesis of *E. coli* O157 phage with 68 kb genome. a** Illustration of O157_vB phage synthesis. DNA fragments with overlapping regions were amplified from phage genome and assembled in vitro. Assembled products were electroporated into *E. coli* HST08. Phages were eventually rebooted inside bacteria and propagated by infecting *E. coli* O157 added to samples before plating. **b** Illustration of insertion sites of 11-amino acid peptide (HiBiT) tag and DNA fragments used. PCR primers containing HiBiT sequence were used to generate O157_vB phages with HiBiT tags at C-terminal of gp37 (minor capsid protein) or gp64 (lytic enzyme). Eight fragments were prepared for each sample. Scale bars represent 1 kb. **c** Evaluation of lytic activities of natural and HiBiT-tagged O157_vB phages against 18 *E. coli* strains.

O157 antigen as a receptor. PP01, another phage that infects *E. coli* O157, not only infected numerous O157 strains but also some non-O157 strains, such as O145, O5, and O172. PP01 uses OmpC as a receptor[51], which may explain its ability to infect certain non-O157 strains. In contrast, O157_vB tagged with HiBiT specifically detected *E. coli* O157. Although O157_vB without the HiBiT tag showed minimal luminescence, this was probably due to the reaction between the added LgBiT protein and the cell lysate before measurement[61].

Additionally, the detection sensitivity, in terms of luminescence levels, varied depending on the position of the tag within the phage and the protein to which the HiBiT tag was fused. To the best of our knowledge, evaluating tagged phages based on their detection sensitivity and considering the location of the peptide tag or reporter protein within the phage are novel aspects of this study. O157_vB, which carries a tag on gp27, exhibited the highest sensitivity in both the ATCC and clinical strains of *E. coli* O157. Although we did not examine the expression levels or structural details of each protein, given that the capsid and tail genes are typically highly expressed[62], we presume that gp27 was sufficiently expressed for detection and that the HiBiT tag did not cause critical structural damage. Moreover, we believe the HiBiT tag was exposed on the outer side of gp27, making detection easier. Previous studies have added reporter proteins to the capsid

proteins of phages[63]. However, we could not synthesize O157_vB carrying the HiBiT tag on two major capsid proteins (gp41 and gp42). Although inserting the tag into a minor capsid protein (gp37) was possible, it exhibited relatively low luminescence. These results indicate that capsid proteins may not always be the optimal choice for incorporating tags or reporter genes. When applying HiBiT-tagged phage-based detection to other phages, the appropriate tag insertion sites differ depending on the existing proteins and their expression levels. Although O157_vB with a tag on gp27 had a higher luminescence, gp64, a lytic enzyme (endolysin), was one of the most suitable target proteins in this study. Phage-encoded lytic enzymes can be useful HiBiT tag targets, rather than using structural proteins or proteins involved in DNA replication.

Although O157_vB may use the O157 antigen as a receptor, the infection efficiency was low in certain *E. coli* O157 strains. Possible reasons for this include low receptor expression levels, which hindered O157_vB attachment, and the presence of defense systems against O157_vB in some *E. coli* O157 strains. Specifically, all tested *E. coli* O157 strains belonging to clade 3 possessed DarTG and Zorya type I systems, which might impede infection by O157_vB and PP01. The study of defense systems and phage infectivity is rapidly advancing worldwide; however, few reports have focused specifically on *E. coli* O157. Comparing the infectivity of O157_vB

**Article**

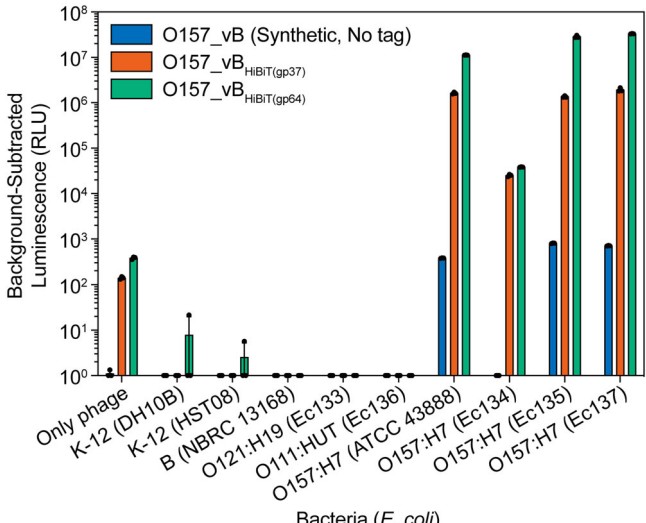

**Fig. 4 | Detection of *E. coli* O157 using HiBiT-tagged O157_vB.** *E. coli* O157 detection assay using synthetic O157_vB phages. 4 *E. coli* O157 strains and 5 non-O157 *E. coli* strains were tested. Luminescence of phage/bacterial cultures was measured after 2 h of incubation. Multiplicity of infection (MOI) at beginning was 1. Lower limit was set at $10^0$ relative light units (RLU). Outcomes are represented as individual replicates ($n$ = 3) and mean with standard deviation (SD).

**Table 1 | O157_vB proteins selected for HiBiT tag addition**

|   | Target protein | Size | Protein function | Number of plaques (avg., $n$ = 2) | HiBiT tag sequence |
|---|---|---|---|---|---|
| 0 | No tag | – | – | 332 | – |
| 1 | gp19 | 208aa | Unknown | 56.5 | Yes |
| 2 | gp20 | 100aa | Unknown | 151 | Yes |
| 3 | gp27 | 779aa | Tail fiber protein | 49 | Yes |
| 4 | gp32 | 76aa | Unknown | 288.5 | Yes |
| 5 | gp33 | 486aa | Terminase large subunit | 0 | – |
| 6 | gp35 | 138aa | Unknown | 336 | Yes |
| 7 | gp36 | 791aa | Portal protein | 249 | Yes |
| 8 | gp37 | 283aa | Minor capsid protein | 215 | Yes |
| 9 | gp41 | 205aa | Major capsid protein | 0 | – |
| 10 | gp42 | 379aa | Capsid and scaffold protein | 0 | – |
| 11 | gp48 | 512aa | Tail sheath protein | 0.5 | No |
| 12 | gp55 | 186aa | Tail fiber protein | 59 | Yes |
| 13 | gp57 | 726aa | Soluble lytic murein transglycosylase | 0 | – |
| 14 | gp62 | 912aa | Putative structural protein | 34 | No |
| 15 | gp64 | 214aa | Lytic enzyme (endolysin) | 224.5 | Yes |
| 16 | gp69 | 171aa | Lysis system i-spanin | 0 | – |
| 17 | gp75 | 965aa | DNA polymerase III alpha subunit | 2 | Yes |
| 18 | gp85 | 108aa | Unknown | 147.5 | Yes |
| 19 | gp86 | 360aa | Zinc-independent RepB primase | 129.5 | Yes |
| 20 | gp91 | 556aa | DNA primase | 75.5 | Yes |

and PP01 and examining different clades could help elucidate important defense systems specific to *E. coli* O157. If these defense systems are found to be the cause of low phage infectivity, a search for inhibitors can be conducted. Once identified, the inhibitors can be incorporated into phages to synthesize phages with higher infectivity, as previously described[64], increasing the sensitivity of *E. coli* O157 detection.

Although O157_vB with HiBiT on gp27 demonstrated the ability to detect various *E. coli* O157 clinical isolates, we did not investigate its detectability in food or clinical samples. As impurities such as other organisms, molecules, or phages in the samples may potentially interfere with the phage infection or luminescence signal, it is essential to verify the detection conditions and thresholds when using phages for these samples. For example, preparing dilution series of food or clinical samples prior to co-incubation with the tagged phages, or dilution series of culture prior to measuring luminescence, would minimize interference. Furthermore, enriching samples in growth media before phage addition could enhance detection in food or clinical samples, as demonstrated in prior studies[20,29]. Adjusting samples to optimal incubation conditions, such as pH, or incorporating measures to counteract interference, if found, would facilitate phage infection, thus preventing loss of detection sensitivity.

Furthermore, the phage-synthetic and synthetic phage-based detection methods described in this study have the following limitations: phage synthesis using in vitro genome assembly is currently difficult to implement for phages with large genomes of more than 100 kb.

Phage-based detection requires phage infection; therefore, if bacteria evolve and become resistant to phages, the phages will be unable to detect them.

Although we did not examine how much toxin would be released, due to the overproduction and secretion of Shiga toxin encoded on phages during lysis, phage-based detection methods should not be used in vivo for detecting STEC strains, and should only be used in restricted areas for in vitro STEC detection. In addition, concerns have been raised about the possibility that the bacterial SOS response could increase horizontal gene transfer and induce Stx phages[65]. To address these concerns, synthetic phages containing toxin inhibitors[66–68] or repressors for Stx phages could be used to detect STEC while suppressing toxin or Stx phage release.

This detection method also relies on luminescence emission from HiBiT tags on synthetic phages, which necessitates luminometers.

Because *E. coli* O157 is detected indirectly through HiBiT-tagged phage infection, luminescence levels calculated from working curves may differ from experimental data at low or high bacteria concentrations, making it difficult to calculate the detection limit.

Finally, the engineered phage exclusively targets *E. coli* O157 and does not detect non-O157 STECs or other pathogens, due to its high specificity for O157 antigen. Therefore, to detect pathogens other than *E. coli* O157, suitable phages must first be isolated before being used for detection, which can be challenging.

To assess the utility of HiBiT-tagged phage-based detection of *E. coli* O157, some aspects such as preparation, time required for detection, accuracy, or accessibility were compared with those for conventional or other phage-based methods and are outlined in Supplementary Table 6. The HiBiT-tagged phage-based detection method developed in this study outperforms culture-based methods in terms of detection time. Unlike PCR-based detection, it allows for easy and straightforward detection and distinguishes viable cells from dead cells. Additionally, it provides high accuracy compared with that for phage particle adsorption or fluorescent-labeled phage-based detection methods and is easier to perform than reporter phage-based detection. These rapid and accurate *E. coli* O157 detection methods could be expanded to other pathogenic bacteria, potentially enabling the simultaneous monitoring and detection of various foodborne pathogens and pathogenic bacteria. Lastly, the phage synthesis method we described can be used in synthetic phage therapy for drug-resistant bacteria and refractory infections. Its short synthesis time also makes it useful for detecting emerging bacterial infections.

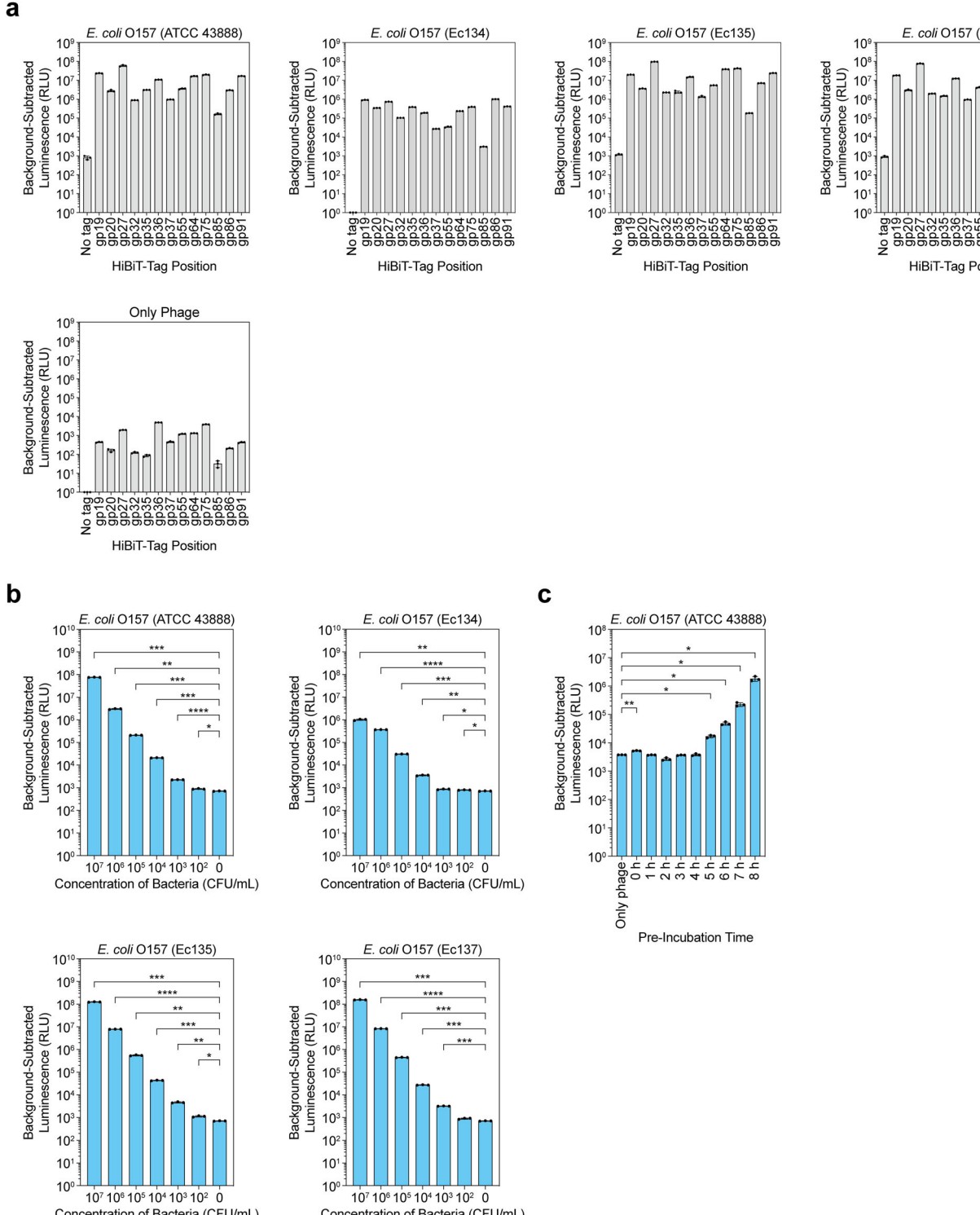

**Fig. 5 | Various O157_vB phages with HiBiT on different proteins. a** *E. coli* O157 detection assay using 13 O157_vB phages with HiBiT tag. Detection sensitivity of phages was compared using Shiga toxin-deficient ATCC strain (ATCC 43888) and clinical isolates (Ec134, Ec135, and Ec137). Phages and bacteria (MOI = 1) were incubated for 2 h, and luminescence of cultures was measured. **b** *E. coli* O157 detection assays for low concentrations. O157_vB$_{HiBiT(gp27)}$ was incubated with tenfold serially diluted bacteria for 2 h, and luminescence of cultures was measured for 7 different concentrations. **c** Detection assay for *E. coli* O157 at starting concentration of 1 CFU/ml. Luminescence of cultures was measured following pre-incubation of 1 CFU/ml *E. coli* O157 (every hour for up to 8 h) and subsequent 2-h incubation with O157_vB$_{HiBiT(gp27)}$. Three technical replicates were performed for each detection assay. Lower limit was set at $10^0$ RLU. Results are shown as each replicate and mean with SD. **b**, **c** Outcomes were analyzed using one-way ANOVA with Brown–Forsythe and Welch tests. Mean of each group was compared with mean of phage-only groups (0 CFU/ml bacteria). Significance is indicated as *$p \leq 0.05$, **$p \leq 0.01$, ***$p \leq 0.001$, or ****$p \leq 0.0001$.

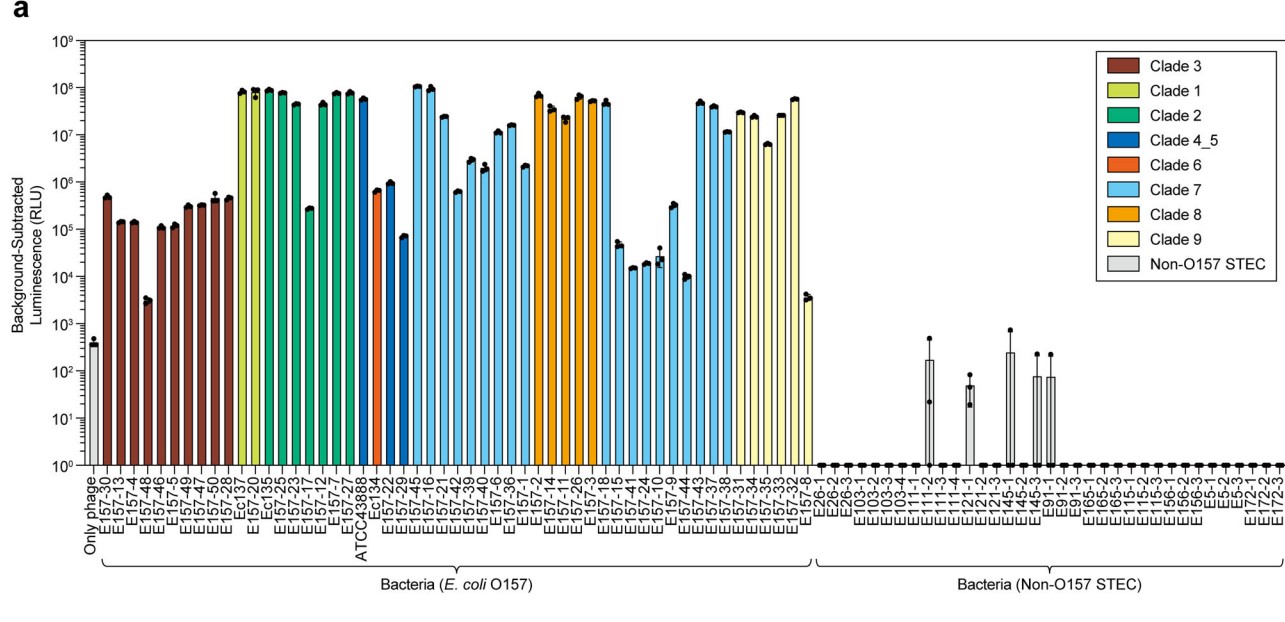

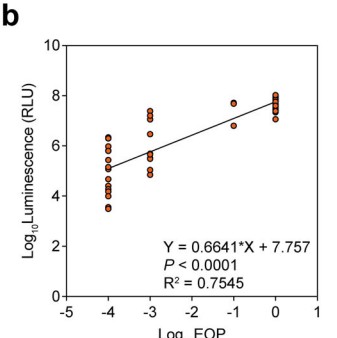

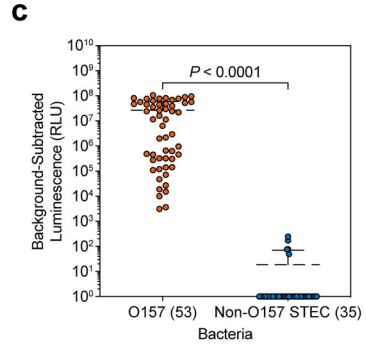

**Fig. 6 | Detection of *E. coli* O157 clinical isolates. a** Detection assays of 53 O157 and 35 non-O157 STEC isolates. Each clinical isolate was incubated with O157_vB$_{HiBiT(gp27)}$ for 2 h at initial MOI of 1, and luminescence of cultures was measured. Lower limit was set at $10^0$ RLU. Results are shown as individual replicates ($n = 3$) and mean with SD. *E. coli* O157 clinical isolates were colored differently depending on their clades. **b** Relationship between O157_vB infectivity and detection sensitivity. EOP values of O157_vB on 53 O157 STEC clinical isolates calculated in Fig. 1c, and luminescence levels are shown as points representing average of three replicates. Influence of EOP on luminescence levels was analyzed using simple linear regression. **c** Specificity of O157_vB detection. Luminescence levels were compared between 53 O157 and 35 non-O157 STEC isolates. Results are displayed as points representing average of three replicates and mean with SD. Outcomes were analyzed using Welch's *t* test for two-group comparisons.

## Methods

### Phage and bacterial strains

The phage and bacterial strains used in this study are described in Supplementary Table 7. *E. coli* O157 Ec134, Ec135, and Ec137 clinical isolates are from the Research Institute for Microbial Diseases, Osaka University, Japan, and the other *E. coli* O157 strains are from the National Institute of Infectious Diseases, Japan. *E. coli* O157 *rfbE*-deficient mutants were previously created, with ATCC 43888 Δ*rfbE::cat* being constructed with pKD3 instead of pKD4[56].

### Phylogenetic analysis of phages

A proteomic tree of the O157_vB genome sequence was generated using ViPTree[69] (v.3.3) under the default settings. Nine *E. coli* phages (T1–T7, SP15, and PP01) were selected for comparison, and a regenerated proteomic tree of their sequences was constructed.

### Spot assay for phage lytic activity

A single bacterial colony was inoculated into 3 ml of Luria-Bertani (LB) media and incubated overnight at 37 °C with shaking at 200 rpm. We added 100 µl of the overnight culture to 3 ml of LB top agar (LTA) containing 0.5% molten agarose and 1 mM CaCl$_2$. The LTA was maintained at 56 °C. We

then poured the mixture onto LB agar plates (15 ml). For large LB plates (30 ml), 10 ml of LTA with 300 µl of the overnight culture was used. The plates were spotted with 10-fold serial phage dilutions. After overnight incubation at 37 °C, the plaques were counted. The EOP against various *E. coli* strains was calculated compared with that of the natural host. *E. coli* K-12 strain DH10B was used as a natural host for *E. coli* phages T1–T7 and SP15, and *E. coli* O157 strain ATCC 43888 was used for PP01 and O157_vB. When propagating an *rfbE*-deficient ATCC 43888 harboring an RfbE-expressing plasmid (ATCC 43888 Δ*rfbE::cat*/pGEM-T-Easy-*rfbE*+), we added 50 µg/ml ampicillin to LB media and 100 µg/ml ampicillin to LB plates.

### Extraction of phage genomes

The phages were first propagated in liquids. We added 100 µl of host bacterial overnight culture to 20 ml of LB media and incubated them at 37 °C with shaking at 200 rpm until the optical density reached 0.1. Subsequently, $10^6$–$10^7$ PFU/ml of phages and 1 mM CaCl$_2$ were added to the culture and incubated with the bacteria. After a 2–3 h incubation, the culture was centrifuged at 8000 × *g* for 15 min at 20 °C, and the supernatant was collected. The supernatant was passed through 0.45-µm filters (Nippon Genetics Co, Ltd., Tokyo, Japan) and reacted with nucleases (1 U/ml DNase I [Nippon

Gene Co., Ltd., Tokyo, Japan], 10 µg/ml RNase A [Nippon Gene Co., Ltd.]) for 1 h at 37 °C. Subsequently, the filtered supernatant was mixed with the same volume of polyethylene glycol solution (10% [w/v] PEG 8000 [MP Biomedicals, Inc., Irvine, CA, USA], 1 M NaCl, 5 mM Tris-HCl [pH 7.5], and 5 mM MgSO$_4$) and incubated overnight at 4 °C. The solution was centrifuged at 10,000 × $g$ for 20 min at 4 °C, and the supernatant was removed. The pellet was resuspended in the remaining solution and centrifuged at 10,000 × $g$ for 15 min at 4 °C. After removing the supernatant, the pellet was dissolved with 500 µl of saline magnesium (SM) buffer (100 mM NaCl, 50 mM Tris-HCl [pH 7.5], 7 mM MgSO4, and 0.01% [w/v] gelatin).

The genomic DNA was then extracted using the phenol–chloroform extraction method. Phenol:chloroform:isoamyl alcohol (25:24:1 [v/v], pH 7.9) was added at the same volume as the that of the samples and mixed well, followed by centrifugation at 10,000 × $g$ for 3 min at 20 °C. The supernatant was collected, and these steps were repeated twice or thrice until the middle layer, containing denatured proteins, between the upper aqueous phase and the lower organic phase became invisible. An equal amount of chloroform:isoamyl alcohol (24:1 [v/v]) was added to the supernatant and thoroughly mixed. After centrifugation at 12,000 × $g$ for 10 min at 20 °C, the supernatant was transferred to new tubes and mixed with the same volume of isopropanol. The supernatant was removed following centrifugation at 10,000 × $g$ for 10 min at 4 °C, and the pellet was washed with 70% ethanol. After centrifugation at 10,000 rpm for 10 min at 4 °C, the supernatant was discarded. Finally, the genomic DNAs were eluted using 50 µl of Tris-EDTA buffer (10 mM Tris-HCl [pH 8.0], 1 mM EDTA).

## In vitro phage DNA assembly

DNA fragments were prepared from the phage genomes using PCR (Supplementary Tables 8 and 9). The primers were designed to create overlapping regions of adjacent fragments. To synthesize phages with the 11-amino-acid HiBiT tag, primers were used, including the HiBiT sequence (VSGWRLFKKIS) at the C-terminal of the targeted proteins. The PCR products were purified using gel extraction methods with FastGene Gel/PCR Extraction Kit (NIPPON Genetics Co, Ltd.). When highly concentrated DNA fragments were required, multiple gels containing the same fragment were passed through the same spin column. Subsequently, the DNA fragments were assembled using overlapping regions. Equimolar DNA fragments (20–50 fmol) and NEBuilder HiFi DNA Assembly (New England Biolabs, Inc., Ipswich, MA, USA) were mixed in 30 µl of reaction mixtures and incubated for 3 h at 50 °C. We prepared samples with water instead of the assembly enzymes as negative controls. The assembly reaction was confirmed via electrophoresis. To clean up the assembled samples, drop dialysis was performed for 20 min using MF-Millipore membrane filters (pore size: 0.025 µm) (Merck KGaA, Darmstadt, Germany).

## Rebooting of phages from assembled genomes

The assembled genomes (10 µl) were electroporated into $E.$ $coli$ HST08 Premium Electro-Cells (Takara Bio Inc., Shiga, Japan) with ELEPO21 (Nepa Gene Co., Ltd., Chiba, Japan; poring pulse [voltage: 1500 V, pulse length: 2.5 ms, pulse interval: 50 ms, number of pulses: 1, polarity: +], transfer pulse [voltage: 150 V, pulse length: 50 ms, pulse interval: 50 ms, number of pulses: 5, polarity: +/−]). After electroporation, samples were added to 1 ml SOC medium (Takara Bio Inc.) and incubated at 37 °C for 20 to 60 min with shaking at 200 rpm. The samples (500 µl to 1 ml) and $E.$ $coli$ O157 strain ATCC 43888 overnight cultures (200–300 µl) were added to 3 ml of LTA. They were poured onto LB plates and incubated at 37 °C overnight. The plaques were counted, and synthetic phages were kept in 100 µl of SM buffer at 4 °C for the following experiments.

## Optimization of phage synthesis

To optimize the phage synthesis methods, we considered several conditions regarding DNA purification, the length of overlapping regions, and the amount of DNA fragment input for assembly using T7 phages. Three sets of DNA fragments were prepared and purified using gel purification, column-based DNA purification using FastGene Gel/PCR Extraction Kit, and no

purification. The DNA assembly and subsequent steps were performed using the same methods. Subsequently, to determine the appropriate overlap length for assembly, the overlapping regions were designed to be 20, 40, 60, 80, or 100 bp. Five pairs of DNA fragments were amplified using PCR. All fragments were subjected to gel purification. To investigate the appropriate amount of DNA fragments to input, equimolar amounts (5, 25, 50, 75, or 100 fmol) of purified DNA fragments were mixed with NEBuilder HiFi DNA Assembly in 30 µl reaction mixture and incubated for 3 h at 50 °C. For all T7 syntheses, the assembled samples were diluted 1–100 times with water and electroporated into $E.$ $coli$ HST08 Premium Electro-Cells. After incubation at 37 °C for 20 min at 200 rpm, all sample volumes were added to 3 ml of LTA and poured onto LB agar plates. The plaques were counted after a 5-h incubation at 37 °C, and the efficiency of T7 phage synthesis under different conditions was compared.

## Confirmation of HiBiT tagging

To confirm the insertion and location of the HiBiT tags, we used the obtained O157_vB plaques as PCR templates and amplified the DNA around the HiBiT sequence (Supplementary Tables 8 and 10). The PCR products were purified using column-based methods (FastGene Gel/PCR Extraction Kit), and Sanger sequencing was performed.

## $E.$ $coli$ O157 detection assay

Except for the detection of low concentrations of $E.$ $coli$ O157, all detection assays, including the detection of clinical isolates, followed the procedure described below. Bacteria (~$10^9$ CFU/ml), cultured overnight, were diluted 100 times in LB media containing 1 mM CaCl$_2$, and synthetic O157_vB phages were diluted to $10^8$ PFU/ml in SM buffer. The bacterial strains and O157_vB phage variants used for each assay are described in Supplementary Table 7. In 96-well plates, 180 µl of the bacteria ($10^7$ CFU/ml in reaction) and 20 µl of the phages ($10^7$ PFU/ml in reaction, multiplicity of infection [MOI] at 1) were incubated at 37 °C at 600 rpm using BioTek LogPhase 600 Microbiology Reader (Agilent Technologies, Inc., Santa Clara, CA, USA). The optical density (OD600) was measured at 10-min intervals. Five wells contained only LB media with 1 mM CaCl$_2$ for the background subtraction. After a 2-h incubation, HiBiT tags on the synthetic O157_vB phages were detected using the Nano-Glo HiBiT Lytic Detection System (Promega Corporation, Madison, WI, USA). We added 50 µl culture and 50 µl Nano-Glo HiBiT Lytic Reagent to 96-well white plates and incubated the mixture for 10 min at 500 rpm with the light blocked. Luminescence was measured using GloMax Explorer Multimode Microplate Reader (integration time: 2 s) (Promega Corporation). Three technical replicate experiments were performed for each assay.

## Detection of low concentrations of $E.$ $coli$ O157

Overnight cultures of $E.$ $coli$ O157 (ATCC 43888, Ec134, Ec135, and Ec137) (~$10^9$ CFU/ml) were serially diluted tenfold in LB media containing 1 mM CaCl$_2$, and 180 µl of each dilution was mixed with 20 µl of $10^8$ PFU/ml O157_vB$_{HiBiT(gp27)}$ suspended in SM buffer ($10^7$ PFU/ml in reaction). After 2-h incubation at 37 °C and 600 rpm, 50 µl of the culture was mixed with 50 µl of Nano-Glo HiBiT Lytic Reagent, and luminescence was measured. Three technical replicates were conducted.

An overnight culture of $E.$ $coli$ O157 ATCC 43888 was serially diluted and used to inoculate 10 ml of LB media with 1 mM CaCl$_2$ at ~1 CFU/ml. The concentration was estimated from enumerating the culture prior to dilution. The culture was then incubated at 37 °C at 600 rpm for 8 h, with 3 samples taken every hour for analysis with the phage. After incubating 9.9 ml of the culture with 100 µl of $10^9$ PFU/ml O157_vB$_{HiBiT(gp27)}$ suspended in SM buffer ($10^7$ PFU/ml in reaction) for 2 h, luminescence was measured as described above. Three technical replicates were performed.

## Extraction of bacterial genomes

NucleoSpin DNA Stool (Takara Bio Inc.) was used to extract the genome of the $E.$ $coli$ O157 strain ATCC 43888, and the MonoFas gDNA Bacteria Extraction Kit VII (ANIMOS Inc., Saitama, Japan) was used for $E.$ $coli$ O157

clinical isolates Ec134, Ec135, and Ec137. Genomic DNAs from the other O157 and non-O157 STEC clinical isolates were extracted using the DNeasy Blood & Tissue Kit (Qiagen GmbH, Hilden, Germany).

## Whole-genome sequencing

For the O157_vB phages and *E. coli* O157 strains (ATCC 43888, Ec134, Ec135, and Ec137), genomic DNA libraries were prepared using NEBNext Ultra II DNA Library Prep Kit for Illumina (New England Biolabs, Inc.) and sequenced using a NovaSeq6000 System (150-mer paired-end) (Illumina, Inc., San Diego, CA, USA). Sequence raw reads were assembled using Shovill (https://github.com/tseemann/shovill) (v.1.0.9 or v.1.1.0) under the default settings. Subsequently, the proteins were annotated using Prokka[70] (v.1.14.6) or DFAST[71] (v.1.6.0) from the assembled contigs. Genomic DNA libraries of the other O157 and non-O157 STEC clinical isolates were prepared using the QIAseq FX DNA Library Kit (Qiagen). The pooled libraries were subjected to multiplexed paired-end sequencing (150-mer paired-end sequencing) using HiSeq X (Illumina, Inc.). The short reads were assembled using SPAdes[72,73] (v.3.13.0) with the "--careful" option. Clades and SG were determined in silico according to the method described by Manning et al.[11]. The core genome single nucleotide polymorphism (SNP)-based phylogenetic relationships of STEC O157 isolates were inferred by an in-house pipeline[74,75] using BactSNP[76] (v.1.1.0) with the genome of STEC O157 strain Sakai (GenBank: BA000007.3) as a reference. The repetitive regions longer than 50 bp were detected by MUMmer[77] (v.4.0.0) (nucmer, repeat-match, and exact-tandems functions) and removed for further analyses, as were prophage regions. The recombinogenic regions were detected by Gubbins[78] (v.2.4.1) and removed. The resultant concatenated SNP sequences were used for further analyses. Model selection and construction for a phylogenetic tree by the maximum likelihood method were performed using ModelTest-NG[79] and RAxML-NG[80] (v.0.9.0) with 1,000 bootstrap replicates.

## Analysis of phage-resistant mutants

To predict the host receptor for O157_vB, we obtained O157_vB-resistant *E. coli* O157 mutants and analyzed their genomes. We incubated 100 μl *E. coli* O157 (ATCC 43888) overnight culture in 10 ml LB media containing 1 mM $CaCl_2$ at 37 °C for 1 h with shaking at 200 rpm. We added $10^6$–$10^7$ PFU/ml phages to the culture and incubated the mixture at 37 °C at 200 rpm. After overnight incubation, the culture was centrifuged at $8000 \times g$ for 10 min at 20 °C, and the supernatant was discarded. The pellet was washed with 10 ml SM buffer and centrifuged at $8000 \times g$ for 10 min at 20 °C. The pellet was resuspended in 10 ml LB medium containing 1 mM $CaCl_2$, and their 10-fold serial dilutions were spotted onto an LB agar plate, which was incubated overnight at 37 °C. Single colonies were incubated in 3 ml of LB medium overnight at 37 °C at 200 rpm, and spot assays of O157_vB were performed to confirm that *E. coli* O157 mutants became resistant to O157_vB. After the whole-genome sequencing of the phage-resistant mutants, core SNPs were analyzed using Snippy (https://github.com/tseemann/snippy) (v.4.6.0). The mutations found in coding sequences are listed in Supplementary Table 1.

## Search for phage defense systems in *E. coli* O157 clinical isolates

We used DefenseFinder[81,82] (v.1.2.2) to examine the presence of phage defense systems in the *E. coli* O157 clinical isolates used in this study.

## Statistics and reproducibility

All statistical tests were performed using GraphPad Prism 9 (v.9.5.1; GraphPad Software Inc., San Diego, CA, USA). Welch's *t* test was used for two-group comparisons, and one-way analysis of variance (ANOVA) with the Brown–Forsythe and Welch tests was used for multi-group comparisons. To assess the detection sensitivity of various HiBiT-tagged O157_vB phages, each *E. coli* O157 strain was tested using 14 phages, including O157_vB without tag (Fig. 5a and Supplementary Table 2). For the detection of low concentrations of *E. coli* O157 using O157_vB$_{HiBiT(gp27)}$, 7 different

concentrations of each *E. coli* O157 strain were compared (Fig. 5b and Supplementary Table 3). To determine the required time for pre-incubation to detect 1 CFU/ml *E. coli* O157 using O157_vB$_{HiBiT(gp27)}$, 9 different time points and only phage samples were compared (Fig. 5c and Supplementary Table 4). To determine the detectability of O157 and non-O157 STEC isolates using O157_vB$_{HiBiT(gp27)}$, 53 O157 and 35 non-O157 STEC samples were compared (Fig. 6c). Each experiment included three technical replicates. In addition, simple linear regression was used to generate working curves and investigate the relationship between phage infectivity and detection sensitivity. Working curves for *E. coli* O157 detection by O157_vB$_{HiBiT(gp27)}$ were created using data from 5 different concentrations ($10^3$–$10^7$ CFU/ml) for each *E. coli* O157 strain (Supplementary Fig. 8), and the relationship between EOP and luminescence levels of 53 *E. coli* O157 strains was examined using O157_vB (Fig. 6b). Each experiment included three technical replicates. *p* values ≤ 0.05 were considered to indicate statistical significance.

## Reporting summary

Further information on research design is available in the Nature Portfolio Reporting Summary linked to this article.

## Data availability

The genome sequence of *E. coli* O157 phage vB_Eco4M-7 is available in GenBank as accession number MN176217[34]. Its whole genome was re-sequenced (GenBank: PP596836), and the genomes were compared using pairwise sequence alignment (Smith–Waterman) in SnapGene (v.7.0.2; GSL Biotech LLC, San Diego, CA, USA). Supplementary Table 11 summarizes a newly discovered repeat region, the DNA sequence of a highly mutated gene (locus tag: vBEco4M7_62), and other mutations. The data that support the findings of this study are openly available in Figshare (https://doi.org/10.6084/m9.figshare.25256308).

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

## Acknowledgements

The O157 phage vB_Eco4M-7 was a gift from Dr. Grzegorz Węgrzyn, Department of Molecular Biology, Faculty of Biology, University of Gdańsk, Wita Stwosza 59, 80-308, Gdańsk, Poland. PP01 was provided by Dr. Yasunori Tanji. We thank the Hibiki Research Group for Clinical Microbiology for providing cephem- and quinolone-resistant *E. coli* strains and carbapenemase-producing *E. coli* strains. We thank Dr. Teppei Sasahara and Dr. Dai Akine of Jichi Medical University for kindly providing the *E. coli* strains. The RIMD EHEC strains were provided by the Research Institute for Microbial Diseases (RIMD) of Osaka University. This work was supported by the Japan Agency for Medical Research and Development (grant No. 21fk0108496 and 21wm0325022 to K. Kiga and 21gm1610002 and 22ae0121045 to L.C. and K. Kiga), JSPS KAKENHI (grant No. 22J22026 to A.T. and 18K15149 to K. Kiga). The funders had no role in the study design, data collection and analysis, decision to publish, or preparation of the manuscript.

## Author contributions

Azumi Tamura: conceptualization, methodology, investigation, data curation, formal analysis, visualization, writing—original draft, and funding acquisition. Aa Haeruman Azam: conceptualization, methodology, resources, and writing—original draft. Tomohiro Nakamura: investigation, data curation, formal analysis, and writing—review & editing. Kenichi Lee: investigation, data curation, formal analysis, resources, and writing—review & editing. Sunao Iyoda: investigation, data curation, resources, and writing—review & editing. Kohei Kondo: data curation, formal analysis, and writing—review & editing. Shinjiro Ojima: resources and writing—review & editing. Kotaro Chihara: resources and writing—review & editing. Wakana Yamashita: resources and writing—review & editing. Longzhu Cui: resources, writing—review & editing, and funding acquisition. Yukihiro Akeda: resources and writing—review & editing. Koichi Watashi: resources and writing—review & editing. Yoshimasa Takahashi: resources and writing—review & editing. Hiroshi Yotsuyanagi: resources and writing—review & editing. Kotaro Kiga: conceptualization, methodology, writing—original draft, funding acquisition, project administration, and supervision.

## Competing interests

The authors declare the following competing interests: A.T., A.H.A., Y.T., K.W. and K. Kiga are co-inventors on a patent pending submitted by the National Institute of Infectious Diseases based on the results reported in this paper.
