## [peer review file · Communications Biology]

Reviewers' comments:

Reviewer #1 (Remarks to the Author):

The manuscript "Synthetic phage-based approach for ultrasensitive and highly specific detection of Escherichia coli O157:H7" COMMSBIO-23-4319 by Tamura et al., showcases the development of a phage-based detection system to accurately detect Escherichia coli serotype O157:H7 isolates. The authors have engineered series of phages with luminescence peptide that enables rapid the detection 88 clinical isolates of O157 antigen, from which 53 were Shiga toxin-producing and 35 non-producers. They authors focus their study on validating the performance of the O157_vBHiBiT phage and the optimisation by cloning the HiBiT peptide in different proteins that are essential for the lytic cycle of their phage variants. In particular, the version with the tail fibre protein (gp27) fused with the tag showed the highest level of luminescence after 2 hour post infection.

I congratulate the authors for their new technology and efforts to challenge conventional detection strategies and the current standards for the identification of this particular pathogen. Overall, I find the manuscript well written and easy to follow, their proof of concept is evidence of the great potential that phages possess. I think there is novelty on their application, but the manuscript falls short with the discussion and scope of their application.

Major Comments:

While the technology developed is exciting, I would like to encourage the authors to revise their manuscript and add include the following aspects in their manuscript/discussion. They focus a considerable part to describing how to engineer the O157_vBHiBiT phage variants and then on the performance for detecting the isolates, but lack suggestions on its implementation and discussion on the potential barriers and limitations that they application could face

It is important to understand the proposed methodology for its use in clinics. I am raising such concern because handling STEC is particularly dangerous and when lysing such strains, Shiga toxin is highly secreted. Therefore, I would encourage the authors to explain the reader why their technology will benefit from detecting STECs and discuss two aspects that present risk: i) the overproduction of the phage-encoded Shiga toxin when lysing cells, and ii) the potential of increasing horizontal gene transfer by inducing the SOS-response and potential release of the STX phages. (see Mühldorfer, I. et al. Regulation of the Shiga-like toxin II operon in Escherichia coli. Infect. 1996)

The technology has clear advantages over existing ones, but there is no comparison or further discussion on how the O157_vBHiBiT phages outperform other technologies (including phage-based and potentially PCR-based). In the introduction, the authors mentioned the use of labelled phages and phage-inducible chromosomal islands to detect E. coli, but they do not compare their technology

to those referenced. What is the benefit of detection upon lysis compared to detecting viable cells? How does luminescence detection compares to fluorescent-based? And could other tags be implemented the same way?

Finally, they briefly highlight that anti-viral defences could present a limitation when identifying the pathogen. How to overcome this? Could a similar approach be used where lysis is not required? (e.g. detection upon attachment).

Minor comment:

Figure 2 could be add as supplementary, as the rebooting of phages from extracted DNA or overlapping PCRs has been shown in previous works.

Reviewer #2 (Remarks to the Author):

In this manuscript, the authors successfully synthesized the O157 antigen-specific phage VB_Eco4M-7 with a genome. HiBiT detection tags were then inserted into the phage genome, and the developed detection method demonstrated good specificity. However, the detection described in the manuscript is preliminary. A major revision is needed before re-consideration. Major comments:

1. Compared with the method of isolating phages in other papers, what are the advantages of this manuscript's phage synthesis method?
2. Please explain in detail why inserting HiBiT tags into gp27 genome works best.
3. The authors mention that attaching the HiBiT tag to the C-terminal of the tail fiber protein gp27 significantly improves the detection sensitivity. However, in the results of Figure 4b, the effect of inserting the tag into gp27 protein is not significantly different from that of other proteins such as gp64 and gp75. What is the basis for significantly improving the detection sensitivity?
4. Please detail the analytical results such as working curve, detection limit, detection range, recovery rate, coefficient of variation. A comparison with other works of E. coli O157: H7 assay in a table is necessary.
5. What is the concentration of different strains in the specificity test?
6. Specific peptide fusion phage proteins isolated from phage display had been reported in detecting pathogens or viruses. In the Introduction, related work should be cited to reflect recent advancements:

<https://doi.org/10.1021/acsami.9b23101>

<https://doi.org/10.1016/j.bios.2016.03.075>

<https://doi.org/10.1039/C7NR06633C>

<https://doi.org/10.1021/acs.analchem.2c01988>

<https://doi.org/10.1021/acs.analchem.3c02821>

Reviewer #3 (Remarks to the Author):

The manuscript is generally well written and organised which made it a pleasure to review.

Abstract

Line 21. I suggest you replace “food poisoning” with “food-borne infection”. Food poisoning can be used to refer to food-borne intoxication or food-borne infection, and since STEC O157 cause food-borne infections.

Lines 23, 29, and throughout the manuscript the abbreviation O157:H7 is used for the bacterium. I recommend that you instead use the term E. coli O157:H7 or E. coli O157 to avoid confusion between the bacterium and the antigen.

I recommend a revision of the abstract as it does not describe the study presented accurately. The impression given by the current abstract is that this study was focused on developing a method for the clinical detection of STEC O157:H7. Instead the paper describes the creation of synthetic phage that uses a specific cell surface antigen as a receptor. The ability of the phage to infect diverse isolates of E. coli with the receptor (O157 antigen) is demonstrated. A relatively minor part of the paper the application of the phage with a luminosity system to detect phage infection, but this amounts to proof of principle experiments as the these experiments were conducted against pure cultures of E. coli O157:H7 instead of food or clinical samples.

Introduction

Line 52. The term culture independent for molecular or serological methods of detection is misleading as in most applications they require cultural enrichment of the sample prior to analysis. An exception being certain types of clinical samples in which the organisms can be expected at high numbers. In comparing the time to result for various methods you may want to note the PCR based methods for STEC can potentially give a negative result in less than 16 hours, with presumptive positive requiring confirmation by isolation.

Line 70. How is the phage vB_Eco4M-7 expected to be applied to benefit “food-protection”?

Line 93. I recommend that you revise this paragraph to emphasize the novel aspect of this study which is the advances in phage engineering. Then inform the reader that though there are multiple potential applications for a targeted phage you conducted a proof of principle study on applying the phage for detection of bacteria expressing the target receptor.

I note that the phage and detection system described appear to be specific for E. coli expressing the O157 antigen. This group can be expected to include E. coli O157 with other H antigens, Shiga toxin negative E. coli O157:H7 and exclude strains of E. coli O157 with the rough phenotypes (without O-oligosaccharide expression). Consequently, I recommend that you refer to the phage infecting and detecting E. coli O157, rather than O157:H7.

Methods

Line 353 Correct to: “E. coli O157 detection assay”

Line 354. What was the volume of bacterial cells and the volume of phage inoculum? What media or buffer were they suspended in?

Line 365 correct to “Detection of low concentrations of E. coli O157”

Lines 366 to 374. The structure of this experiment is not completely clear. Am I correct in understanding that two experiments were performed with E. coli O157 strain ATCC 43888:

A. An overnight culture of E. coli O157 was serially diluted ten fold (in media or buffer?) and aliquots (what volume?) of those dilutions were mixed with 10^7 PFU/mL of the phage (is that the concentration of in the phage inoculum or in reaction with the bacteria? And what volume). Incubated for 2 h and the 50 μ L of culture was mixed with 50 μ L of Nano-Glo and luminescence measured.

B. An overnight culture of E. coli O157 was serially diluted and used to inoculate 10 mL of LB broth at approximately 1 CFU/mL (that approximate is important because I know no routine method for diluting and inoculating bacterial cells with that level of precision, do you have counts of the culture prior dilution to make an estimate from?). The culture was then incubated for 8 hours with aliquots taken at intervals (what were they?) for analysis with the phage. Phage analysis involving a further 2 h incubation (volumes/ media?) before mixing of 50 μ L of culture with 50 μ L of Nano-Glo and luminescence measurement.

These experimental details are important in understanding the sensitivity (Limit of detection) of the analysis and reporting it correctly.

Please review the sections “E. coli O157 detection assay” and “Detection of low concentrations of E. coli O157” and ensure they provide complete details. Including the number of replicate experiments, and the estimated CFU estimated from enumerating the culture prior to dilution . It is also unclear which specific O157_vB phage variants (different proteins tags) these experiments were conducted with.

Results

Lines 175 to 179. The description of the sensitivity of the assay is confusing. In part due to the incomplete methods provided for these experiments.

A 10^2 CFU/mL culture is described as giving a positive result after 2h incubation, was that incubation period prior tot exposure to the phage. And in describing the sensitivity the CFU per reaction volume (200 μ L?) provides a more complete description.

Similarly there are problems with the description of the detection of 1 CFU/mL with a 7 h incubation. Was this:

- a. 7 h incubation with the phage;
- b. 7 h incubation plus 2 h incubation with the phage;
- c. 5 h incubation plus 2 h incubation with the phage.

If case c. then assuming a an average cell density of 1 cell/mL, the initial 10 mL culture would contain 10 cells and these would double every 20 min (doi: 10.1128/JB.01368-07). Thus in 5 hours 327,680 cells, let us assume 300,000 to account for lag phase at the beginning of the enrichment, that is 30,000 CFU/mL. If the culture in the reaction volume is 200 μ L then the LOD is 6000 CFU. That a very different sensitivity than that implied by a statement that 1 CFU/mL can be detected.

Discussion

Lines 237 to 243. This section on the detection experiments could benefit from some expansion. If the food and clinical samples were not tested because interference from other organisms/molecules/phages is expected how would this system be developed beyond a proof of concept? Any discussion of applications should include and account for an accurate description of the sensitivity of the analysis and time to result. As described in this paper though innovate the phage detection is not faster or more sensitive than PCR, so what advantages could it offer?

Figures and Tables

These are very complex and contain a great deal of information, with each figure containing multiple graphs, illustrations and tables, which would typically be presented individually. Additionally, presenting this information as small images and tables on crowded pages makes it difficult to read and interpret. Some of this information would seem better suited for the supplementary material.

Some such as photographs of phage plaque plates and electrophoresis gels provided minimal information to the reader and could largely be removed. I highly recommend that the Figures be revised with the aim of concision, to present the most relevant information in the simplest format.

References

The formatting of reference is not always consistent and the name of organisms in titles should be in italics.

Reviewers' comments:

Reviewer #1 (Remarks to the Author):

The manuscript “Synthetic phage-based approach for ultrasensitive and highly specific detection of Escherichia coli O157:H7” COMMSBIO-23-4319 by Tamura et al., showcases the development of a phage-based detection system to accurately detect Escherichia coli serotype O157:H7 isolates. The authors have engineered series of phages with luminescence peptide that enables rapid the detection 88 clinical isolates of O157 antigen, from which 53 were Shiga toxin-producing and 35 non-producers. They authors focus their study on validating the performance of the O157_vBHiBiT phage and the optimisation by cloning the HiBiT peptide in different proteins that are essential for the lytic cycle of their phage variants. In particular, the version with the tail fibre protein (gp27) fused with the tag showed the highest level of luminescence after 2 hour post infection.

I congratulate the authors for their new technology and efforts to challenge conventional detection strategies and the current standards for the identification of this particular pathogen. Overall, I find the manuscript well written and easy to follow, their proof of concept is evidence of the great potential that phages possess. I think there is novelty on their application, but the manuscript falls short with the discussion and scope of their application.

Dear Reviewer 1:

Thank you for your thorough review of our manuscript and for providing valuable feedback to help us improve it. We have revised the manuscript based on your comments, focusing mainly on the Discussion section. The changes to the text are shown in yellow highlights in the manuscript file. Our point-by-point responses to your comments are given below.

Major Comments:

Comment 1: While the technology developed is exciting, I would like to encourage the authors to revise their manuscript and add include the following aspects in their manuscript/discussion. They focus a considerable part to describing how to engineer the O157_vBHiBiT phage variants and then on the performance for detecting the isolates,

but lack suggestions on its implementation and discussion on the potential barriers and limitations that they application could face.

Response 1:

In response to the comment, we discussed the use of the HiBiT tag on other phages (**page 6, lines 247-251**):

When applying HiBiT-tagged phage-based detection to other phages, the appropriate tag insertion sites may differ depending on the existing proteins and their expression levels. Although O157_vB with a tag on gp27 had a higher luminescence, gp64, a lytic enzyme (endolysin), was one of the most suitable target proteins in this study. Lytic enzymes, rather than structural proteins or proteins involved in DNA replication, can be useful HiBiT tag targets.

We also listed the limitations of phage synthesis and phage-based detection methods (**page 6, lines 270-289**):

Furthermore, the phage-synthetic and synthetic phage-based detection methods described in this study have the following limitations: Phage synthesis using in vitro genome assembly is currently difficult to implement for phages with large genomes of more than 100 kb. Phage-based detection requires phage infection; therefore, if bacteria evolve and become resistant to phages, the phages will be unable to detect them. Although we did not examine how much toxin would be released, due to the overproduction and secretion of Shiga toxin encoded on phages during lysis, phage-based detection methods should not be used in vivo for detecting STEC strains, and should only be used in restricted areas for in vitro STEC detection. In addition, concerns have been raised about the possibility that the bacterial SOS response could increase horizontal gene transfer and induce Stx phages⁶⁵. To address these concerns, synthetic phages containing toxin inhibitors or repressors for Stx phages could be used to detect STEC while suppressing toxin or Stx phage release. This detection method also relies on luminescence emission from HiBiT tags on synthetic phages, which necessitates luminometers. Because *E. coli* O157 is detected indirectly through HiBiT-tagged phage infection, luminescence levels calculated from working curves may differ from experimental data at low or high bacteria concentrations,

making it difficult to calculate the detection limit. Finally, the engineered phage exclusively targets *E. coli* O157 and does not detect non-O157 STECs or other pathogens, due to its high specificity for O157 antigen. Therefore, to detect pathogens other than *E. coli* O157, suitable phages must first be isolated before being used for detection, which can be challenging.

Comment 2: It is important to understand the proposed methodology for its use in clinics. I am raising such concern because handling STEC is particularly dangerous and when lysing such strains, Shiga toxin is highly secreted. Therefore, I would encourage the authors to explain the reader why their technology will benefit from detecting STECs and discuss two aspects that present risk: i) the overproduction of the phage-encoded Shiga toxin when lysing cells, and ii) the potential of increasing horizontal gene transfer by inducing the SOS-response and potential release of the STX phages. (see Mühldorfer, I. et al. Regulation of the Shiga-like toxin II operon in Escherichia coli. Infect. 1996)

Response 2:

Thank you for raising important concerns about the use of synthetic phage-based *E. coli* O157 detection in clinics. This method cannot be used for in vivo detection and should only be used in limited areas for in vitro detection because, as you mentioned, Shiga toxin can be overproduced and secreted during cell lysis. We agree that horizontal gene transfer and Stx phage release induced by the bacterial SOS response are potential risks. To address these, synthetic phages containing toxin inhibitors or Stx phage repressors could be used for detection (**page 6, lines 275-281**) (See above.)

We also made a list to compare the features of some *E. coli* O157 detection methods (**Supplementary Table 6**). HiBiT-tagged phage-based detection developed in this study outperforms culture-based methods in terms of detection time. Furthermore, it allows for easy and simple detection and can distinguish viable cells from dead cells when compared to PCR-based detection (**pages 6-7, lines 290-297**).

To assess the utility of HiBiT-tagged phage-based detection of *E. coli* O157, some aspects such as preparation, time required for detection, accuracy, or accessibility were compared with those for conventional or other phage-based methods and are outlined in Supplementary Table 6. The HiBiT-tagged phage-based detection method developed in this study outperforms culture-based methods in terms of detection time. Unlike PCR-based detection, it allows for easy and straightforward detection and distinguishes viable cells

from dead cells. Additionally, it provides high accuracy compared with that for phage particle adsorption or fluorescent-labeled phage-based detection methods and is easier to perform than reporter phage-based detection.

Comment 3: The technology has clear advantages over existing ones, but there is no comparison or further discussion on how the O157_vBHiBiT phages outperform other technologies (including phage-based and potentially PCR-based). In the introduction, the authors mentioned the use of labelled phages and phage-inducible chromosomal islands to detect *E. coli*, but they do not compare their technology to those referenced. What is the benefit of detection upon lysis compared to detecting viable cells? How does luminescence detection compares to fluorescent-based? And could other tags be implemented the same way?

Response 3:

In response to the comment, we compared the *E. coli* O157 detection method developed in this study to conventional or other phage-based detection methods and summarized the results (**Supplementary Table 6**). Compared with phage particle adsorption-based detection, which does not require lysis, detection upon lysis allows for more sensitive detection and distinguishing between viable and dead cells. The HiBiT (11-amino acid peptide) tag used in this study can produce strong and quantifiable luminescence after binding to LgBiT, derived from NanoLuc, and it is short enough to insert directly into PCR primers and easily incorporate into phages (**page 5, lines 217-220**), whereas both luminescence and fluorescent-based methods can be used to detect pathogens. Other tags, such as those used in western blotting, are not applicable to this detection method.

Typically, phage genomes are limited by the gene length that can be inserted into the capsid, making it challenging to increase phage genome length. However, introducing the HiBiT tag into the *E. coli* O157 phage was easily achieved. The HiBiT tag, which comprises only 11 amino acids, was of sufficient length to fit within the phage capsid.

Comment 4: Finally, they briefly highlight that anti-viral defences could present a limitation when identifying the pathogen. How to overcome this? Could a similar approach be used where lysis is not required? (e.g. detection upon attachment).

Response 4:

To overcome anti-phage defense systems, searches can be made for inhibitors against them. Once identified, the inhibitors can be incorporated into phages and used to synthesize phages with higher infectivity, potentially allowing for the detection of *E. coli* O157 strains that were previously less detectable (**page 6, lines 260-263**).

If these defense systems are found to be the cause of low phage infectivity, a search for inhibitors can be conducted. Once identified, the inhibitors can be incorporated into phages to synthesize phages with higher infectivity, as previously described⁶⁴, increasing the sensitivity of *E. coli* O157 detection.

Anti-phage defense systems that counteract phage infection after attachment are unlikely to affect the sensitivity of phage particle adsorption-based detection or other methods that do not require cell lysis. However, these detection methods only use phage proteins rather than whole phages, and there is no propagation or reporter gene amplification, resulting in reduced sensitivity (**Supplementary Table 6**).

Minor comment:

Comment 5: Figure 2 could be added as supplementary, as the rebooting of phages from extracted DNA or overlapping PCRs has been shown in previous works.

Response 5:

Thank you for this suggestion. We moved the procedure illustration (previous Fig. 2a) to Fig. 3, along with the figures of HiBiT-tagged phage synthesis (previous Figs. 3a and c) (**Fig. 3a**), removed the illustration of fragments overlapping, and moved the remaining images to the supplementary figures (**Supplementary Figs. 4b and c**). We would like to include the procedure illustration in the main panel because rebooting phages that infect *E. coli* O157 requires a different procedure: incubation with *E. coli* O157 without chloroform treatment.

Reviewer #2 (Remarks to the Author):

In this manuscript, the authors successfully synthesized the O157 antigen-specific phage VB_Eco4M-7 with a genome. HiBiT detection tags were then inserted into the phage genome, and the developed detection method demonstrated good specificity. However, the detection described in the manuscript is preliminary. A major revision is needed before re-consideration.

Dear Reviewer 2:

Thank you for taking time to review this manuscript and for your valuable comments, which have helped us to improve the manuscript. In responding to your feedback, we focused primarily on revising the Results and Discussion sections. The changes to the text are shown in yellow highlights in the manuscript file. Our point-by-point responses to your comments are given below.

Major comments:

1. Compared with the method of isolating phages in other papers, what are the advantages of this manuscript's phage synthesis method?

Response 1:

Compared with traditional phage genome engineering methods such as homologous recombination, the phage synthesis method used in this study allows for easier and more flexible manipulation of phage genomes, as demonstrated by the construction of various types of HiBiT-tagged phages (**page 5, lines 213-216**):

In vitro assembly-based synthesis offers greater flexibility than recombination-based techniques in manipulating phage genomes. This enables the straightforward construction of diverse tagged phages and facilitates the alteration of phage host range⁴⁴. This method also improves synthesis efficiency and eliminates the need for phage selection.

This method has also been shown to be effective for easily switching phage host ranges (Mitsunaka, S. *et al.* Synthetic engineering and biological containment of bacteriophages. *Proc Natl Acad Sci U S A*, 2022). Furthermore, because of its high synthesis efficiency, this method does not require counter-selection.

2. Please explain in detail why inserting HiBiT tags into gp27 genome works best.

Response 2:

In response to the comment, we added explanations to the discussion (**page 6, lines 239-242**):

Given that the capsid and tail genes are typically highly expressed⁶², we presume that gp27 was sufficiently expressed for detection and that the HiBiT tag did not cause significant structural damage. Moreover, we believe the HiBiT tag was exposed on the outer side of gp27, making detection easier.

3. The authors mention that attaching the HiBiT tag to the C-terminal of the tail fiber protein gp27 significantly improves the detection sensitivity. However, in the results of Figure 4b, the effect of inserting the tag into gp27 protein is not significantly different from that of other proteins such as gp64 and gp75. What is the basis for significantly improving the detection sensitivity?

Response 3:

We chose the O157_vB phage with a HiBiT tag on the tail fiber protein gp27 for the subsequent detection assays because it exhibited the highest luminescence of the 20 tagged phages (**pages 4-5, lines 182-185**):

The amount of observed luminescence varied among the 13 phages, and O157_vB with a HiBiT tag at gp27 (O157_vB_{HiBiT(gp27)}), a tail fiber protein, showed the highest level of luminescence after a 2-h incubation with *E. coli* O157 (Fig. 5a). Statistical analysis using multi-group comparisons confirmed significant differences (**Supplementary Table 2**).

4. Please detail the analytical results such as working curve, detection limit, detection range, recovery rate, coefficient of variation. A comparison with other works of *E. coli* O157: H7 assay in a table is necessary.

Response 4:

Thank you for your comment. Using luminescence data from the detection of low concentrations of *E. coli* O157 (10^3 – 10^7 CFU/mL), we created working curves for *E. coli*

O157 detection by O157_vB with a HiBiT tag at gp27 (Supplementary Fig. 8). Please see the figures below.

We found that *E. coli* O157 concentrations of 10³ CFU/mL or higher correlated with luminescence levels, whereas those of 10² CFU/mL did not exhibit such correlation (page 5, lines 189-191).

Because *E. coli* O157 is detected indirectly via infection by HiBiT-tagged phages and amplification of HiBiT-tagged proteins with phage propagation, luminescence levels calculated from working curves may differ from experimental data at low or high bacterial concentrations, making calculating the detection limit or other related parameters difficult (page 6, lines 283-285).

5. What is the concentration of different strains in the specificity test?

Response 5:

We updated the methodology to include more detailed explanations. In the specificity test, all strains were adjusted to approximately 10⁷ CFU/mL in the reaction (page 9, lines 406-413):

Except for the detection of low concentrations of *E. coli* O157, all detection assays, including the detection of clinical isolates, followed the procedure described below. Bacteria (approximately 10⁹ CFU/mL), cultured overnight,

were diluted 100 times in LB media containing 1 mM CaCl₂, and synthetic O157_vB phages were diluted to 10⁸ PFU/mL in SM buffer. The bacterial strains and O157_vB phage variants used for each assay are described in Supplementary Table 7. In 96-well plates, 180 μL of the bacteria (10⁷ CFU/mL in reaction) and 20 μL of the phages (10⁷ PFU/mL in reaction, multiplicity of infection [MOI] at 1) were incubated at 37 °C at 600 rpm using BioTek LogPhase 600 Microbiology Reader (Agilent Technologies, Inc., Santa Clara, CA, USA).

6. Specific peptide fusion phage proteins isolated from phage display had been reported in detecting pathogens or viruses. In the Introduction, related work should be cited to reflect recent advancements:

<https://doi.org/10.1021/acsami.9b23101>

<https://doi.org/10.1016/j.bios.2016.03.075>

<https://doi.org/10.1039/C7NR06633C>

<https://doi.org/10.1021/acs.analchem.2c01988>

<https://doi.org/10.1021/acs.analchem.3c02821>

Response 6:

Thank you for your suggestion and the references. In the Introduction, we included a representative reference to pathogen detection using specific fusion phage proteins (Liu, P. *et al.* Colorimetric Assay of Bacterial Pathogens Based on Co(3)O(4) Magnetic Nanozymes Conjugated with Specific Fusion Phage Proteins and Magnetophoretic Chromatography. *ACS Appl Mater Interfaces*, 2020) (**page 2, lines 63-64**):

Moreover, pathogen detection has been facilitated using specific fusion phage proteins selected from phage display libraries²⁶.

Reviewer #3 (Remarks to the Author):

The manuscript is generally well written and organised which made it a pleasure to review.

Dear Reviewer 3:

Thank you for your thorough review of our paper and for the extensive feedback that you provided. We have revised the manuscript based on your comments. In responding to your comments, we specifically revised the Abstract, Introduction, and Methods sections to make them more understandable. The changes to the text are shown in yellow highlights in the manuscript file. Our point-by-point responses to your comments are given below.

Abstract

Comment 1: Line 21. I suggest you replace “food poisoning” with “food-borne infection”. Food poisoning can be used to refer to food-borne intoxication or food-borne infection, and since STEC O157 cause food-borne infections.

Response 1:

According to your suggestion, we changed “food poisoning” to “foodborne infection” (page 2, line 21).

Comment 2: Lines 23, 29, and throughout the manuscript the abbreviation O157:H7 is used for the bacterium. I recommend that you instead use the term *E. coli* O157:H7 or *E. coli* O157 to avoid confusion between the bacterium and the antigen.

Response 2:

Thank you for pointing this out. We agree that the abbreviation O157:H7 was misleading and have used *E. coli* O157 throughout the revised manuscript instead, because H antigens were not tested for some clinical isolates and the phages used in this study recognize the O157 antigen.

Comment 3: I recommend a revision of the abstract as it does not describe the study presented accurately. The impression given by the current abstract is that this study was focused on developing a method for the clinical detection of STEC O157:H7. Instead the paper describes the creation of synthetic phage that uses a specific cell surface antigen as a receptor. The ability of the phage to infect diverse isolates of *E. coli* with the receptor

(O157 antigen) is demonstrated. A relatively minor part of the paper the application of the phage with a luminosity system to detect phage infection, but this amounts to proof of principle experiments as these experiments were conducted against pure cultures of *E. coli* O157:H7 instead of food or clinical samples.

Response 3:

Thank you for your advice. We revised the abstract to state that we used an in vitro assembly-based synthesis of vB_Eco4M-7, an O157-antigen-specific phage, for the phage-based detection of *E. coli* O157 as a proof of concept, which has the potential to be applied to other pathogenic bacteria for detection or other applications such as phage therapy (**page 2, lines 21-30**). Please see the revised abstract below:

Escherichia coli O157 infection can cause mass foodborne infection, leading to severe disease such as hemolytic-uremic syndrome. Although phage-based detection methods for *E. coli* O157 are being explored, research on their accuracy with clinical isolates is lacking. Here, we describe an in vitro assembly-based synthesis of vB_Eco4M-7, an O157 antigen-specific phage with a 68-kb genome, and its use as a proof of concept for *E. coli* O157 detection. Among 20 possible insertion sites for the detection tag, linking it to the C-terminus of the tail fiber protein, gp27, greatly enhances detection sensitivity. The constructed phage detects all 53 diverse clinical isolates of *E. coli* O157, clearly distinguishing them from 35 clinical isolates of non-O157 Shiga toxin-producing *E. coli*. Our efficient phage synthesis methods can be applied to other pathogenic bacteria for a variety of applications, including phage-based detection and phage therapy.

Introduction

Comment 4: Line 52. The term culture independent for molecular or serological methods of detection is misleading as in most applications they require cultural enrichment of the sample prior to analysis. An exception being certain types of clinical samples in which the organisms can be expected at high numbers. In comparing the time to result for various methods you may want to note the PCR based methods for STEC can potentially give a negative result in less than 16 hours, with presumptive positive requiring confirmation by isolation.

Response 4:

Thank you for pointing out that the term “culture independent” is inappropriate. We agree that this gives the impression that no cultural enrichment is required prior to analysis. We removed the term (**page 2, line 51**).

Comment 5: Line 70. How is the phage vB_Eco4M-7 expected to be applied to benefit “food-protection”?

Response 5:

Thank you for the question. Because of its rapid propagation and ability to lyse various *E. coli* O157 strains, the lytic phage vB_Eco4M-7 is expected to improve food safety measures when used for pathogen detection and neutralization in food (**page 2, lines 72-74**).

Comment 6: Line 93. I recommend that you revise this paragraph to emphasize the novel aspect of this study which is the advances in phage engineering. Then inform the reader that though there are multiple potential applications for a targeted phage you conducted a proof of principle study on applying the phage for detection of bacteria expressing the target receptor.

Response 6:

Thank you for your valuable advice. We rearranged the sentences in the paragraph to emphasize the potential use of an improved phage synthesis method and its implementation as a proof-of-concept for detecting bacteria that express the target receptor. We also included a novel aspect of this study: comparing detection tag insertion sites using synthetic tagged phages (**page 3, lines 98-105**):

Herein, we present an in vitro method for synthesizing an *E. coli* O157-specific phage with a 68-kb genome. The efficient phage-synthetic method enables a targeted phage to be used in a variety of applications, including therapy against drug-resistant bacteria. We conducted a proof of concept study using the phage to detect bacteria expressing its target receptor. Through a comparison of the detection tag insertion sites, we demonstrate a rapid and ultrasensitive *E. coli* O157 detection system based on synthetic tagged phages. The synthetic phage-based detection method could be extended to other pathogenic bacteria, allowing the simultaneous monitoring and detection of various pathogenic bacteria.

Comment 7: I note that the phage and detection system described appear to be specific for *E. coli* expressing the O157 antigen. This group can be expected to include *E. coli* O157 with other H antigens, Shiga toxin negative *E. coli* O157:H7 and exclude strains of *E. coli* O157 with the rough phenotypes (without O-oligosaccharide expression). Consequently, I recommend that you refer to the phage infecting and detecting *E. coli* O157, rather than O157:H7.

Response 7:

Thank you for your recommendation. As previously stated, we replaced O157:H7 with *E. coli* O157 throughout the manuscript to avoid confusion between the bacterium and the antigen and to indicate the *E. coli* that can be infected by the O157-antigen-specific phage used in this study.

Methods

Comment 8: Line 353 Correct to: “*E. coli* O157 detection assay”

Response 8:

We corrected the wording to “*E. coli* O157 detection assay” (page 9, line 405).

Comment 9: Line 354. What was the volume of bacterial cells and the volume of phage inoculum? What media or buffer were they suspended in?

Response 9:

We revised the Methods section to include additional information such as concentration, volume, and media/buffer, as requested.

Page 9, lines 407-409:

Bacteria (approximately 10^9 CFU/mL), cultured overnight, were diluted 100 times in LB media containing 1 mM CaCl₂, and synthetic O157_vB phages were diluted to 10^8 PFU/mL in SM buffer.

Page 9, lines 411-413:

In 96-well plates, 180 μ L of the bacteria (10^7 CFU/mL in reaction) and 20 μ L of the phages (10^7 PFU/mL in reaction, multiplicity of infection [MOI] at 1)

were incubated at 37 °C at 600 rpm using BioTek LogPhase 600 Microbiology Reader (Agilent Technologies, Inc., Santa Clara, CA, USA).

Comment 10: Line 365 correct to “Detection of low concentrations of *E. coli* O157”

Response 10:

We corrected the wording to “Detection of low concentrations of *E. coli* O157.” (page 9, line 423).

Comment 11: Lines 366 to 374. The structure of this experiment is not completely clear. Am I correct in understanding that two experiments were performed with *E. coli* O157 strain ATCC 43888:

A. An overnight culture of *E. coli* O157 was serially diluted ten fold (in media or buffer?) and aliquots (what volume?) of those dilutions were mixed with 10^7 PFU/mL of the phage (is that the concentration of in the phage inoculum or in reaction with the bacteria? And what volume). Incubated for 2 h and the 50 μ L of culture was mixed with 50 μ L of Nano-Glo and luminescence measured.

B. An overnight culture of *E. coli* O157 was serially diluted and used to inoculate 10 mL of LB broth at approximately 1 CFU/mL (that approximate is important because I know no routine method for diluting and inoculating bacterial cells with that level of precision, do you have counts of the culture prior dilution to make an estimate from?). The culture was then incubated for 8 hours with aliquots taken at intervals (what were they?) for analysis with the phage. Phage analysis involving a further 2 h incubation (volumes/media?) before mixing of 50 μ L of culture with 50 μ L of Nano-Glo and luminescence measurement.

These experimental details are important in understanding the sensitivity (Limit of detection) of the analysis and reporting it correctly.

Response 11:

Thank you for your question and important advice. Two experiments were carried out with *E. coli* O157. The first experiment was carried out with the ATCC 43888 strain and clinical isolates Ec134, Ec135, and Ec137, and the second was carried out only with the ATCC 43888 strain. The bacterial concentration was calculated by counting the culture before dilution, so the concentration was only an approximation, as you noted. We revised this section in response to the comment, including providing information about the concentration, volume, and media/buffer (page 9, lines 424-436):

Overnight cultures of *E. coli* O157 (ATCC 43888, Ec134, Ec135, and Ec137) (approximately 10^9 CFU/mL) were serially diluted tenfold in LB media containing 1 mM CaCl₂, and 180 μ L of each dilution was mixed with 20 μ L of 10^8 PFU/mL O157_vB_{HiBiT(gp27)} suspended in SM buffer (10^7 PFU/mL in reaction). After 2-h incubation at 37 °C and 600 rpm, 50 μ L of the culture was mixed with 50 μ L of Nano-Glo HiBiT Lytic Reagent, and luminescence was measured. Three technical replicates were conducted.

An overnight culture of *E. coli* O157 ATCC 43888 was serially diluted and used to inoculate 10 mL of LB media with 1 mM CaCl₂ at approximately 1 CFU/mL. The concentration was estimated from enumerating the culture prior to dilution. The culture was then incubated at 37 °C at 600 rpm for 8 h, with 3 samples taken every hour for analysis with the phage. After incubating 9.9 mL of the culture with 100 μ L of 10^9 PFU/mL O157_vB_{HiBiT(gp27)} suspended in SM buffer (10^7 PFU/mL in reaction) for 2 h, luminescence was measured as previously described. Three technical replicates were performed.

Comment 12: Please review the sections “*E. coli* O157 detection assay” and “Detection of low concentrations of *E. coli* O157” and ensure they provide complete details. Including the number of replicate experiments, and the estimated CFU estimated from enumerating the culture prior to dilution. It is also unclear which specific O157_vB phage variants (different proteins tags) these experiments were conducted with.

Response 12:

Following the comment, we revised the sections to include complete details such as the number of replicates and estimated CFU based on counting prior to dilution (**page 9, lines 405-436**). Furthermore, we included information about the O157_vB phage variants used in each detection assay (**Supplementary Table 7; page 9, lines 409-411**).

Results

Comment 13: Lines 175 to 179. The description of the sensitivity of the assay is confusing. In part due to the incomplete methods provided for these experiments. A 10^2 CFU/mL culture is described as giving a positive result after 2h incubation, was that incubation period prior tot exposure to the phage. And in describing the sensitivity the CFU per reaction volume (200 μ L?) provides a more complete description.

Similarly there are problems with the description of the detection of 1 CFU/mL with a 7 h incubation. Was this:

- a. 7 h incubation with the phage;
- b. 7 h incubation plus 2 h incubation with the phage;
- c. 5 h incubation plus 2 h incubation with the phage.

If case c. then assuming an average cell density of 1 cell/mL, the initial 10 mL culture would contain 10 cells and these would double every 20 min (doi: 10.1128/JB.01368-07). Thus in 5 hours 327,680 cells, let us assume 300,000 to account for lag phase at the beginning of the enrichment, that is 30,000 CFU/mL. If the culture in the reaction volume is 200 μ L then the LOD is 6000 CFU. That a very different sensitivity than that implied by a statement that 1 CFU/mL can be detected.

Response 13:

Thank you for pointing this out. We apologize for the unclear description. For the first part on detecting low concentrations of *E. coli* O157, serially diluted bacteria were mixed with phages and incubated for 2 hours without bacterial enrichment. We have included the CFU per reaction volume in the revised text (**page 5, lines 186-189**):

We then investigated the detection of small numbers of *E. coli* O157 using O157_vB_{HiBiT(gp27)} (Figs. 5b and c). O157_vB_{HiBiT(gp27)} detected approximately 10² CFU/mL (18 CFU/200 μ L) *E. coli* O157 of a Shiga-toxin-deficient ATCC strain (ATCC 43888) and clinical isolates Ec134 and Ec135 after a 2-h co-incubation (Fig. 5b, Supplementary Table 3).

The result of the second part was “c. 5 h incubation plus 2 h incubation with the phage”. We understand that we cannot state that 1 CFU/mL can be detected due to the pre-enrichment of bacteria. We revised the wording to avoid stating that 1 CFU/mL was detected, instead explaining that *E. coli* O157 was detected with adequate pre-incubation, even when the starting concentration was as low as 1 CFU/mL (**page 5, lines 191-193**):

Moreover, after a 5-h incubation of 1 CFU/mL (equivalent to 10 CFU/10 mL), co-incubation of *E. coli* O157 ATCC 43888 with O157_vB_{HiBiT(gp27)} for 2 h led to successful detection (Fig. 5c, Supplementary Table 4).

Discussion

Comment 14: Lines 237 to 243. This section on the detection experiments could benefit from some expansion. If the food and clinical samples were not tested because interference from other organisms/molecules/phages is expected how would this system be developed beyond a proof of concept? Any discussion of applications should include and account for an accurate description of the sensitivity of the analysis and time to result. As described in this paper though innovate the phage detection is not faster or more sensitive than PCR, so what advantages could it offer?

Response 14:

Thank you for your comments. We expanded the discussion to include the limitations and evaluation of the phage synthesis and detection method developed in this study (**pages 6-7, lines 264-302**).

When analyzing food or clinical samples, diluting the samples before adding tagged phages, or diluting the culture before measuring luminescence, would reduce the impact of impurities in these samples, advancing this detection beyond proof-of-concept. In addition, we compared this HiBiT-tagged phage-based *E. coli* O157 detection method to conventional or other phage-based detection methods in some ways, which we outline in **Supplementary Table 6**.

This detection method is easier and simpler than PCR-based detection, and it can distinguish between viable and dead cells (**page 6-7, lines 294-295**):

Unlike PCR-based detection, it allows for easy and straightforward detection and distinguishes viable cells from dead cells.

Figures and Tables

Comment 15: These are very complex and contain a great deal of information, with each figure containing multiple graphs, illustrations and tables, which would typically be presented individually. Additionally, presenting this information as small images and tables on crowded pages makes it difficult to read and interpret. Some of this information would seem better suited for the supplementary material. Some such as photographs of phage plaque plates and electrophoresis gels provided minimal information to the reader

and could largely be removed. I highly recommend that the Figures be revised with the aim of concision, to present the most relevant information in the simplest format.

Response 15:

Thank you for the suggestion. In response to your comment, we rearranged the figures. First, we combined data from spot assays examining the lytic activity of phages O157_vB and PP01 against STEC clinical isolates (previous Fig. 1c) and the phylogenetic tree of STEC O157 strains (previous Fig. 1d) into a single figure (**Fig. 1c**). Please see the figure below:

In addition, we made the data regarding the prediction of the O157_vB host receptor (previous Figs. 1e and f) independent figures, and changed the abbreviation to *E. coli* O157 based on the comments (**Figs. 2a and b**). Please see the figures below.

a

E. coli O157	Mutation	Gene	Product	Source	
O157_vB-resistant	A→C	Val133Gly	wbdO	Glycosyltransferase	This study
	G→T	Ala83Asp	wbdN	Glycosyltransferase	This study
	A→C	Leu196Arg	wbdN	Glycosyltransferase	This study
	G→A	Arg265Cys	wecA	UDP-N-acetylglucosamine--undecaprenyl-phosphate N-acetylglucosamine phosphotransferase	This study
	G→GC	Ala106fs	wzy	O157 family O-antigen polymerase	This study
PP01-resistant	C→T	Gln76Ter	ompC	Porin OmpC	Azam et al. [50]
	A→T	Arg143Ter	–	Glycosyltransferase	[50]
SP15-resistant	C→T	Trp511Ter	fhuA	Ferrichrome porin FhuA	[50]
	G→A	Pro696Leu	–	DEAD/DEAH box helicase	[50]

b
Next, we moved the illustration of O157_vB phage synthesis (previous Fig. 2a) to Fig. 3, along with the figures of HiBiT-tagged phage synthesis (previous Figs. 3a and c) (**Fig. 3a**), and the plate and agarose gel images to supplementary information (**Supplementary Figs. 4b and c, and 5b**).

Furthermore, we separated the data from the *E. coli* O157 detection assay using synthetic phages (previous Fig. 3d) into an independent figure (**Fig. 4**).

Finally, we changed the previous Fig. 4; information about target proteins for HiBiT tags is listed in Table 1 and relocated the other figures, with the abbreviation changed to *E. coli* O157 (**Figs. 5a, b, and c**). Please see the revised figures below.

References

Comment 16: The formatting of reference is not always consistent and the name of organisms in titles should be in italics.

Response 16:

Thank you for the comment. We have formatted the references according to the *Nature* referencing style and have italicized the names of organisms in titles.

REVIEWERS' COMMENTS:

Reviewer #1 (Remarks to the Author):

The authors have greatly improved the manuscript by discussing further their results and making changes to aid the reader with the rationale of their study and their perspective.

I appreciate that they have listed both advantages and limitations of the HiBiT tag on other phages and that they have also added a Supplementary Table comparing the current detection methods for *E. coli* O157.

I have no major comments to add and I am happy to recommend their manuscript for publication. I express my enthusiasm and hopes to see this technology in food products.

Minor comments:

Line 214. Change "This enables the straightforward construction of..." to "This enables a straightforward construction."

Line 222. Change "its very high electroporation efficiency" to "its high electroporation efficiency"

Line 223. Change "subsequently adding" to "co-culture with *E. coli* O157 as the phage host".

Line 244. Delete Furthermore and start with Although...

Line 248. Delete may as you have shown this in the results.

Line 250. Change to "Phage-encoded lytic enzymes can be useful HiBiT tag targets, rather than using structural proteins or proteins involved in DNA replication."

Line 300. Change Furthermore to Lastly

Reviewer #2 (Remarks to the Author):

The authors polished the paper greatly based on reviewers' suggestion. It is recommended to accept the paper in the current version.

Reviewer #3 (Remarks to the Author):

The manuscript "Synthetic phage-based approach for ultrasensitive and highly specific detection of *Escherichia coli* O157" describes the creation of synthetic bacteria phage which targets the O157-antigen of *E. coli* as a receptor. The application of this phage for the detection of *E. coli* O157 cells was investigated by modifying a tail fiber protein to incorporate a fluorescent detection tag.

The revision adequately addresses the issues raised by the original reviewers. A few minor issues remain and should be addressed by the authors in preparing the manuscript for publication.

Abstract

1. Line 21 Suggest “Escherichia coli O157 can cause foodborne outbreaks, with infection leading to severe disease such as hemolytic-uremic syndrome.”

2. Line 23 “... research on their accuracy with clinical isolates ...”

I suggest replacing “accuracy” with “specificity”.

Accuracy is how close a given set of measurements (observations or readings) are to their true value. In qualitative bacteriological testing the ability of a method of analysis to distinguish the target organism from non-target organisms is referred to the Exclusivity or Specificity of the method.

US Food and Drug Administration. Guidelines for the Validation of Analytical Methods for the Detection of Microbial Pathogens in Foods and Feeds, Edition 3.0

<https://www.fda.gov/media/83812/download>

3. Line 25. Suggest. “Linking the detection tag to the C-terminus of the tail fiber protein, gp27 produced the greatest detection sensitivity of the 20 insertions sites tested.”

Introduction

4. Line 44. “In particular, strains belonging to clade 8 of the O157 serotype cause severe symptoms.”

As written the sentence implies that clade 8 STEC O157 are uniquely associated with severe patient outcomes, more accurately citation 11 (Manning et al. 2008) reports that clade 8 has a higher association with severe patient outcomes among the clades included in the study.

5. Line 73. “...using this phage for pathogen detection and neutralization in food...” In the Discussion (line 269-285) the authors propose that the phage interaction with STEC O157 could result in the release of Stx-phage, which if they infect other E. coli could create new STEC strains. How would neutralization of the pathogen in the food be conducted safely?

Results

6. Line 120-121. How many non-O157 STEC strains, and how many different O-types were included in these experiments?

7. Line 196-197. Please indicate how many strains of STEC O157, the STEC O157 clades represented, and the diversity numbers of non-O157 STEC.

Discussion

8. Line 254. Delete “sufficient”

9. Lines 264 to 289. This paragraph seems to be trying to discuss multiple topics. It would be helpful to the reader if this was rewritten with a separate paragraph on each topic.

10. Line. 264 -270. Diluting the sample would reduce the concentration of inhibitors of phage infection or luminescent signal but would also reduce the sensitivity of phage based detection. How could the method be applied with out loss of sensitivity.

10. Line 280. The authors propose the use of the phage detection with synthetic phages

incorporating elements that suppress Stx toxin expression, or Stx phage replication. Have any such elements been identified? Please provide supporting references.

Methods

11. Line 320. Was molten agar used? What temperature?

12. Line 338. Suggest “Subsequently, the filtered supernatant was mixed with the same volume of ...”

13. Line 348. “... these steps were repeated twice or thrice.” Why was there variation?

Line 380. Please define or describe SM buffer on first use.

REVIEWERS' COMMENTS:

Reviewer #1 (Remarks to the Author):

The authors have greatly improved the manuscript by discussing further their results and making changes to aid the reader with the rational of their study and their prospective.

I appreciate that they have listed both advantages and limitations of the HiBiT tag on other phages and that they have also added a Supplementary Table comparing the current detection methods for E. coli O157.

I have no major comments to add and I am happy to recommend their manuscript for publication. I express my enthusiasm and hopes to see this technology in food products.

Dear Reviewer 1:

We appreciate your thorough second review of our manuscript and the helpful feedback you provided. We have changed the words and sentences according to your comments. The revised text is highlighted in yellow in the manuscript file. Our point-by-point responses to your comments are given below.

Minor comments:

Comment 1: Line 214. Change “This enables the straightforward construction of...” to “This enables a straightforward construction.

Response 1:

We corrected the sentence to “This enables a straightforward construction and facilitates the alteration of phage host range⁴⁴” (page 5, line 218).

Comment 2: Line 222. Change “its very high electroporation efficiency” to “its high electroporation efficiency”

Response 2:

We corrected the wording to “its high electroporation efficiency” (page 5, lines 225-226).

Comment 3: Line 223. Change “subsequently adding” to “co-culture with *E. coli* O157 as the phage host”.

Response 3:

Thank you for your suggestion. We changed the wording to “co-culture with *E. coli* O157 as the phage host” (**page 5, lines 226-227**).

Comment 4: Line 244. Delete Furthermore and start with Although...

Response 4:

We changed to start with “Although” according to your comment (**page 6, line 248**).

Comment 5: Line 248. Delete may as you have shown this in the results.

Response 5:

We deleted “may” from the sentence (**page 6, line 251**).

Comment 6: Line 250. Change to “Phage-encoded lytic enzymes can be useful HiBiT tag targets, rather than using structural proteins or proteins involved in DNA replication.”

Response 6:

Thank you for clarifying the sentence. We changed the sentence to “Phage-encoded lytic enzymes can be useful HiBiT tag targets, rather than using structural proteins or proteins involved in DNA replication” (**page 6, lines 254-255**).

Comment 7: Line 300. Change Furthermore to Lastly

Response 7:

We corrected the wording to “Lastly” (**page 7, line 310**).

Reviewer #3 (Remarks to the Author):

The manuscript “Synthetic phage-based approach for ultrasensitive and highly specific detection of Escherichia coli O157” describes the creation of synthetic bacteria phage which targets the O157-antigen of E. coli as a receptor. The application of this phage for the detection of E. coli O157 cells was investigated by modifying a tail fiber protein to incorporate a fluorescent detection tag.

The revision adequately addresses the issues raised by the original reviewers. A few minor issues remain and should be addressed by the authors in preparing the manuscript for publication.

Dear Reviewer 3:

Thank you for your insightful feedback on our paper. We appreciate the time and effort you have invested in reviewing our work. We have carefully considered your comments and have revised the manuscript accordingly. Changes made to the text are indicated with yellow highlights in the manuscript file. Our point-by-point responses to your comments are given below.

Abstract

Comment 1: Line 21 Suggest “Escherichia coli O157 can cause foodborne outbreaks, with infection leading to severe disease such as hemolytic-uremic syndrome.”

Response 1:

According to your suggestion, we changed the sentence to “*Escherichia coli* O157 can cause foodborne outbreaks, with infection leading to severe disease such as hemolytic-uremic syndrome” (page 2, lines 21-22).

Comment 2: Line 23 “... research on their accuracy with clinical isolates ...”

I suggest replacing “accuracy” with “specificity”.

Accuracy is how close a given set of measurements (observations or readings) are to their true value. In qualitative bacteriological testing the ability of a method of analysis to distinguish the target organism from non-target organisms is referred to the Exclusivity or Specificity of the method.

US Food and Drug Administration. Guidelines for the Validation of Analytical Methods for the Detection of Microbial Pathogens in Foods and Feeds, Edition 3.0

<https://www.fda.gov/media/83812/download>

Response 2:

We appreciate you pointing this out. We agree that “accuracy” is inappropriate in this case. We corrected the wording to “specificity” (**page 2, line 23**).

Comment 3: Line 25. Suggest. “Linking the detection tag to the C-terminus of the tail fiber protein, gp27 produced the greatest detection sensitivity of the 20 insertions sites tested.”

Response 3:

Thank you for your suggestion. We changed the sentence to “Linking the detection tag to the C-terminus of the tail fiber protein, gp27 produces the greatest detection sensitivity of the 20 insertions sites tested” based on your comment (**page 2, lines 25-27**). We use the present tense according to the journal’s style and formatting guide.

Introduction

Comment 4: Line 44. “In particular, strains belonging to clade 8 of the O157 serotype cause severe symptoms.” As written the sentence implies that clade 8 STEC O157 are uniquely associated with severe patient outcomes, more accurately citation 11 (Manning et al. 2008) reports that clade 8 has a higher association with severe patient outcomes among the clades included in the study.

Response 4:

We agree that the previous sentence did not explain the Manning’s findings accurately. We are grateful that you pointed this out. We changed the sentence according to your suggestion (**page 2, lines 43-45**):

In particular, strains belonging to clade 8 of the O157 serotype have a higher association with severe patient outcomes among 9 distinct clades¹¹.

Comment 5: Line 73. “...using this phage for pathogen detection and neutralization in food...” In the Discussion (line 269-285) the authors propose that the phage interaction with STEC O157 could result in the release of Stx-phage, which if they infect other E.

coli could create new STEC strains. How would neutralization of the pathogen in the food be conducted safely?

Response 5:

Thank you for pointing out this important contradiction. Because of the Shiga toxin release, we believe that this phage is more suitable for STEC O157 detection than food disinfection or patient treatment. Therefore, we deleted the wording “neutralization” (page 2, lines 73-74):

Because of its rapid propagation and ability to effectively lyse various *E. coli* O157 strains, using this phage for pathogen detection in food is expected to enhance food safety measures.

Results

Comment 6: Line 120-121. How many non-O157 STEC strains, and how many different O-types were included in these experiments?

Response 6:

We corrected the sentence to include the number of non-O157 STEC strains and O-serotypes used in the experiment (page 3, lines 120-121):

In contrast, O157_vB did not infect any of the 35 non-O157 STEC isolates belonging to 11 different O-serotypes.

Comment 7: Line 196-197. Please indicate how many strains of STEC O157, the STEC O157 clades represented, and the diversity numbers of non-O157 STEC.

Response 7:

We corrected the sentence to include the number of strains/clades/O-serotypes used in the experiment (page 5, lines 199-202):

To assess the usefulness of HiBiT-tagged O157_vB for *E. coli* O157 detection, we next examined the detection of 53 clinical isolates of STEC O157 from 8 diverse clades, as well as 35 clinical isolates of non-O157 STEC across 11 different O-serotypes, using O157_vB_{HiBiT(gp27)} (Fig. 6a).

Discussion

Comment 8: Line 254. Delete “sufficient”

Response 8:

We deleted “sufficient” (page 6, line 258).

Comment 9: Lines 264 to 289. This paragraph seems to be trying to discuss multiple topics. It would be helpful to the reader if this was rewritten with a separate paragraph on each topic.

Response 9:

Thank you for your comment. We separated each topic into paragraphs (pages 6-7, lines 279, 282, 284, 291, 293, 296).

Comment 10: Line. 264 -270. Diluting the sample would reduce the concentration of inhibitors of phage infection or luminescent signal but would also reduce the sensitivity of phage based detection. How could the method be applied with out loss of sensitivity.

Response 10:

Thank you for your question. We agree that diluting the sample would also reduce detection sensitivity. We think that sample enrichment prior to phage addition can be useful for pathogen detection in food or clinical samples, as previous studies have shown. In addition, phage-based detection requires adjusting samples to optimal incubation conditions. We revised this section to include additional methods (page 6, lines 272-278):

For example, preparing dilution series of food or clinical samples prior to co-incubation with the tagged phages, or dilution series of culture prior to measuring luminescence, would minimize interference. Furthermore, enriching samples in growth media before phage addition could enhance detection in food or clinical samples, as demonstrated in prior studies^{20,29}. Adjusting samples to optimal incubation conditions, such as pH, or incorporating measures to counteract interference, if found, would facilitate phage infection, thus preventing loss of detection sensitivity.

Comment 11: Line 280. The authors propose the use of the phage detection with synthetic phages incorporating elements that suppress Stx toxin expression, or Stx phage replication. Have any such elements been identified? Please provide supporting references.

Response 11:

Thank you for the valuable advice. Toxin inhibitors have been identified, as shown below, so we have included supporting references (**page 6, line 289**).

66 Bernedo-Navarro, R. A. *et al.* Peptides derived from phage display libraries as potential neutralizers of Shiga toxin-induced cytotoxicity in vitro and in vivo. *J Appl Microbiol* **116**, 1322-1333 (2014). <https://doi.org:10.1111/jam.12451>

67 Watanabe-Takahashi, M. *et al.* Identification of a peptide motif that potently inhibits two functionally distinct subunits of Shiga toxin. *Commun Biol* **4**, 538 (2021). <https://doi.org:10.1038/s42003-021-02068-3>

68 Nakanishi, K. *et al.* Prevention of Shiga toxin 1-caused colon injury by plant-derived recombinant IgA. *Sci Rep* **12**, 17999 (2022). <https://doi.org:10.1038/s41598-022-22851-4>

Methods

Comment 12: Line 320. Was molten agar used? What temperature?

Response 12:

Molten agarose kept at 56 °C was used in this study. We changed sentences to include the necessary information (**page 7, lines 329-331**):

We added 100 µL of the overnight culture to 3 mL of LB top agar (LTA) containing 0.5% molten agarose and 1 mM CaCl₂. The LTA was maintained at 56 °C. We then poured the mixture onto LB agar plates (15 mL).

Comment 13: Line 338. Suggest “Subsequently, the filtered supernatant was mixed with the same volume of ...”

Response 13:

Thank you for your suggestion. We changed “samples” to “filtered supernatant” according to your comment (**page 7, line 348**).

Comment 14: Line 348. “... these steps were repeated twice or thrice.” Why was there variation?

Response 14:

We revised this sentence to include the detail of phenol–chloroform extraction (**page 8, lines 357-360**):

The supernatant was collected, and these steps were repeated twice or thrice until the middle layer, containing denatured proteins, between the upper aqueous phase and the lower organic phase became invisible.

Comment 15: Line 380. Please define or describe SM buffer on first use.

Response 15:

Thank you for your comment. The term "SM buffer" was not used for the first time in line 392 (previous 380), but it was defined on its first use (page 8, lines 353-354).